# Decadal biomass increment in early secondary succession woody ecosystems is increased by $CO_2$ enrichment

Anthony P. Walker[1], Martin G. De Kauwe[2], Belinda E. Medlyn et al.[#]

Increasing atmospheric $CO_2$ stimulates photosynthesis which can increase net primary production (NPP), but at longer timescales may not necessarily increase plant biomass. Here we analyse the four decade-long $CO_2$-enrichment experiments in woody ecosystems that measured total NPP and biomass. $CO_2$ enrichment increased biomass increment by $1.05 \pm 0.26$ kg C m$^{-2}$ over a full decade, a $29.1 \pm 11.7\%$ stimulation of biomass gain in these early-secondary-succession temperate ecosystems. This response is predictable by combining the $CO_2$ response of NPP ($0.16 \pm 0.03$ kg C m$^{-2}$ y$^{-1}$) and the $CO_2$-independent, linear slope between biomass increment and cumulative NPP ($0.55 \pm 0.17$). An ensemble of terrestrial ecosystem models fail to predict both terms correctly. Allocation to wood was a driver of across-site, and across-model, response variability and together with $CO_2$-independence of biomass retention highlights the value of understanding drivers of wood allocation under ambient conditions to correctly interpret and predict $CO_2$ responses.

f Earth's major terrestrial biomes, forests have the greatest potential to remove atmospheric carbon at decadal time scales due to their relatively high productivity and storage of carbon in long-lived woody structures[1]. Forest soils also contain substantial carbon; however, estimated changes in decadal storage rates in soils in response to increasing $CO_2$ enrichment are smaller[2] than for live vegetation biomass (leaves, wood and roots). Furthermore, biomass responses are the larger component of uncertainty in Earth System model projections of terrestrial carbon sink responses to increasing $CO_2$[3–5]. This uncertainty is due, in large part, to limited predictive understanding of the ecosystem processes that determine the fate of additional carbon that arises from the stimulation of photosynthesis by $CO_2$ enrichment[6,7]. Higher atmospheric $CO_2$ may not stimulate forest net primary production (NPP) if the supply of photosynthetic carbon does not limit NPP at ambient $CO_2$ concentrations[8], as in cases where the availability of other resources (e.g. nitrogen; N) limit NPP[9]. Even if NPP is stimulated by $CO_2$ enrichment, tree biomass may not increase if the additional NPP is allocated to fast-turnover foliage or fine-roots[10] or if tree mortality rates increase[11]. Whether increasing atmospheric $CO_2$ increases plant biomass increment at longer timescales has remained an open question since the early days of $CO_2$ research[12]:

"… the initial effect of elevated $CO_2$ will be to increase NPP in most plant communities. This increase in NPP could be limited or reversed by nutrient availability, herbivory, or successional dynamics […]. However, even without such effects, a critical question is the extent to which the increase in NPP will lead to a substantial increase in plant biomass. Alternatively, increased NPP could simply increase the rate of turnover of leaves or roots without changing plant biomass."

Long-term, ecosystem-scale $CO_2$ enrichment experiments provide the most direct evidence of whether rising atmospheric $CO_2$ may lead to increased forest biomass carbon. Here we synthesise NPP and biomass responses to $CO_2$ from four ecosystem $CO_2$ enrichment experiments. We selected experiments that lasted a decade or longer, were sited in ecosystems dominated by woody plants and that were unmanaged during the experiment, were replicated at the ecosystem scale, and where all components of NPP and biomass were quantified (Table 1).

Our selection criteria restrict the analysis to temperate woody ecosystems, in the early phases of secondary succession. There were two such experiments in deciduous forests: Oak Ridge National Laboratory (ORNL)[13] and Rhinelander[14], and two in evergreen forests: Duke[15] and Kennedy Space Center

(KSC)[16]. All sites were situated in the temperate zone and in the USA; covering several climatic regions, humid continental (Rhinelander) and humid sub-tropical (Duke, KSC and ORNL), and spanning a climatic gradient (mean annual temperature 6.0–22.1 °C; mean annual precipitation 662–1221 mm; Table 1). At these sites, all major components of NPP were measured, site and species-specific allometric relationships were developed to calculate wood biomass at all sites, litterfall was collected at three sites, and root biomass and production were variously measured across sites (see Methods for further discussion of differences in measurements). Three sites were enriched in $CO_2$ using Free Air $CO_2$ Enrichment (FACE) technology on 25–30 m diameter plots of 2–3 replicates per treatment. KSC was enriched in $CO_2$ using 2 m diameter Open Top Chambers (OTC) with eight replicates per treatment. The FACE experiments all increased $CO_2$ by about 50% above ambient while at KSC the increase was about 100%.

Our objectives are threefold: determine whether a decade of $CO_2$ enrichment in woody ecosystems leads to an increase in the vegetation biomass increment ($\Delta C_{veg}$); interpret any observed biomass response through the effects of $CO_2$-enrichment on NPP and carbon allocation; and evaluate the ability of an ensemble of terrestrial ecosystem models, commonly used to predict vegetation responses to $CO_2$, to reproduce the observed responses.

The shifts in resource availability associated with secondary succession[17–19] are likely key factors influencing the $CO_2$ response[18–21]. Therefore we begin the analysis by inferring the successional stage of each site from trends in NPP, leaf area, and fine-root biomass.

To analyse responses to elevated $CO_2$ across all four sites in a unified statistical framework we use linear mixed-effects models, treating site as a random effect. Site as a random effect treats each experiment as a sample drawn from a population (in the statistical sense), allowing the estimation of a population-level fixed effect and its associated uncertainty. This hierarchical mixed-model analysis allows an estimate of the general effect of $CO_2$ in the population of early-secondary-succession, temperate woody ecosystems and the random effects estimate inter-site variability. We use Akaike Information Criterion corrected for finite sample size (AICc) to select the best, most parsimonious, statistical model from a set of candidate models.

A set of equations are derived to decompose the biomass increment response to $CO_2$ and to interpret the empirical parameters of the linear mixed-effects models in the context of carbon allocation (see Methods for more details). We evaluate the ability of a model ensemble to predict decadal biomass responses to $CO_2$ enrichment and identify general areas of model failure where

**Table 1 Experiment site description**

| Site | Forest type | Dominant species | Dominant PFT | Time since disturbance | Climate* | MAT (C) | MAP (mm) | Soil type | Target $CO_2$ (ppm) | $CO_2$ enrichment method |
|------|-------------|------------------|--------------|------------------------|----------|---------|----------|-----------|---------------------|--------------------------|
| Rhinelander | Establishing plantation | Populus tremuloides | Deciduous broadleaf | 1 | Dfa | 6.0 (0.8) | 662 (122) | Alfic Haplorthod | 560 | FACE |
| ORNL | Unmanaged plantation forest | Liquidambar styraciflua | Deciduous broadleaf | 10 | Cfa | 14.8 (0.9) | 1221 (218) | Aquic Hapludult | 565 | FACE |
| Duke | Unmanaged plantation forest | Pinus taeda | Evergreen needleleaf | 13 | Cfa | 14.8 (0.6) | 1081 (168) | Ultic Hapludalf | Ambient + 200 | FACE |
| KSC | Natural woodland, regularly disturbed | Quercus spp | Evergreen broadleaf | 0 | Cfa | 22.1 (0.4) | 1094 (207) | Arenic Haplahumods & Spodic Quartzipsamments | 700 | OTC |

*MAT* mean annual temperature, *MAP* mean annual precipitation
*Köppen-Geiger climate classification: C warm temperate, D continental or snow climates, f fully humid, a hot summer

the whole ensemble fails to reproduce observations. Models are evaluated against the biomass response and the decomposition of the biomass response. Recommendations are provided for future research to help understand and predict vegetation biomass responses to $CO_2$ enrichment.

Results show that a decade of $CO_2$ enrichment stimulates live-biomass increment in early-secondary-succession, temperate woody ecosystems and that the rate of conversion of NPP to biomass was not directly affected by $CO_2$.

## Results

**Analysis of successional stage.** Following Bormann and Likens[18] we interpret the successional stage of these ecosystems using the first three stages of their classification: reorganisation, aggrading and transition. We also interpret these ecosystems (and successional stages) in the context of three coupling states of growth to resource availability described by Körner[19]: expanding, coupled, and uncoupled. Trends in key ecosystem variables—NPP, peak leaf area index (LAI), and fine-root biomass—indicate the dynamics of resource acquisition and space filling in relation to resource acquisition, which in turn are related to secondary-successional stage[18,19]. The reorganisation stage immediately following a disturbance is indicated by an increasing linear trend in NPP[18,22], caused by expanding resource acquisition volumes both above and below ground. Expanding resource acquisition volumes are indicated by increasing trends in LAI and fine-root biomass[17,19]. Aggrading and transition phases are both stages where growth is coupled to light availability and endogenous nutrient cycling[18,19]. We associate the absence of a trend in NPP, LAI and fine-root biomass with the aggrading stage and declining trends with the transition stage[18].

The dynamics of annual NPP, LAI, and fine-root biomass in the ambient $CO_2$ treatment were different across sites (Fig. 1) and these early secondary successional ecosystems do not fit neatly into a single successional phase[18] or resource coupling state[19]. At Rhinelander, the site with the youngest trees, there was a linear increase in NPP matched by increasing LAI and fine-root biomass (Fig. 1a, e, h). These trends indicate that Rhinelander was in the reorganisation stage of succession with expanding resource acquisition volumes.

At ORNL, NPP increased initially, peaked, and then declined in the later years to below the initial NPP (Fig. 1b). Fine-root biomass exhibited a similar trend (Fig. 1i). The peaked trend was not matched by LAI, though there appeared to be a decline in the final years. The sweetgum stand was N limited[23] and N addition alleviated the decline in NPP in ambient $CO_2$[13] suggesting, along with $^{15}N$ data[24], that the forest was experiencing progressive nitrogen limitation under ambient $CO_2$ conditions. We propose that coupled, tightening resource availability and intensifying competition caused the forest at ORNL to undergo a shift from the aggrading to transition phases during the course of the experiment.

NPP at Duke showed strong inter-annual variability but no trend (Fig. 1c). LAI and fine-root biomass showed substantial inter-annual variability, but in contrast with NPP, LAI increased early in the experiment and then saturated, and fine-root biomass increased throughout the experiment (Fig. 1g, j). Duke appears to have been in the expanding resource acquisition phase for light early in the experiment and for below-ground resources during the whole experiment (fine-root biomass saturated in the elevated $CO_2$ treatment with fine-root biomass in the ambient treatment approaching this saturation value by the end of the experiment). However, the lack of a trend in NPP data suggests that the expanding resource acquisition volumes over time were not yielding increased resource acquisition (Fig. 1b). Thus we

conclude that Duke was primarily in a coupled resource state in the aggrading stage of succession.

The strongest inter-annual variability in NPP was seen at KSC with a declining linear trend (Fig. 1d). At KSC, the inter-annual variability of fine-root biomass was strong (Fig. 1k, similar to NPP) and no clear trend was apparent (unlike NPP). The declining NPP trend suggests a rapid response post fire that decreased over time. This rapid post-fire response could have been supported by nutrients released from fire and large below-ground reserves indicated by 5–10-fold higher fine-root biomass than at the other three sites. The very high NPP in the early years of the experiment suggests that growth was uncoupled from resource availability immediately following the disturbance at KSC, with coupling to resources increasing as NPP declines through the experiment. KSC does not fit cleanly into the scheme of Bormann and Likens as the disturbance did not kill the trees and the stand is recovering from live below-ground organs and not from seedling re-establishment.

**Forest responses to $CO_2$ enrichment.** Over the full duration of the experiments, mixed-model analysis revealed that $CO_2$ enrichment (to 550–700 $\mu$mol mol$^{-1}$) increased population-level biomass increment ($\Delta C_{veg}$) $\pm$ SEM by 1.05 $\pm$ 0.26 kg C m$^{-2}$ above a population-level ambient $\Delta C_{veg}$ of 3.62 $\pm$ 1.16 kg C m$^{-2}$, an increase of 29.1 $\pm$ 11.7% (Table 2, model 1; and Supplementary Table 1). Comparison of AICc's provided weak evidence that the random effect of site applied only to the intercept and not the $CO_2$ response (Supplementary Table 1). While the mean $\Delta C_{veg}$ responses at each site showed large variability (Fig. 2), statistical detection of these differences (through random effects on the eCO$_2$ term) was obscured by high within-treatment variability and thus high within-site uncertainty in their mean responses.

Mixed-model analysis also revealed that $CO_2$ enrichment increased population-level mean annual NPP $\pm$ SEM by 0.16 $\pm$ 0.03 kg C m$^{-2}$ y$^{-1}$ above a population-level ambient NPP of 0.72 $\pm$ 0.13 kg C m$^{-2}$ y$^{-1}$ (Table 2, model 2; and Supplementary Table 2). Thus the population-level response to $CO_2$ enrichment was 22.9 $\pm$ 6.1%. As with $\Delta C_{veg}$, the random effects suggested that the absolute response of annual NPP to $CO_2$ enrichment was not statistically different across sites (Table 2 and Supplementary Table 2). Again the detection of site level differences in the $CO_2$ response was obscured by high within-site and within-treatment variability. This variability in NPP was likely driven in large part by heterogeneity in N availability[15,25].

We attempted to explain some of the uncertainty in the $\Delta C_{veg}$ response by adding cumulative NPP (cNPP) to the explanatory model of $\Delta C_{veg}$. This more extensive statistical model of $\Delta C_{veg}$ revealed a strong positive relationship between $\Delta C_{veg}$ and cumulative NPP (cNPP), a relationship that did not include an effect of $CO_2$ enrichment in the best model (Fig. 2; Table 2, model 3; Supplementary Table 3, row 10). The absence of $CO_2$ treatment as a predictor in the best model means that the change in $\Delta C_{veg}$ for a unit change in cNPP (d$\Delta C_{veg}$/dcNPP; hereafter referred to as the biomass retention rate) was maintained across $CO_2$ treatments (at 0.55 $\pm$ 0.17, unitless). Therefore, the $\Delta C_{veg}$ response to $CO_2$ depended on the cNPP response to $CO_2$ constrained by the $CO_2$-independent, biomass retention rate. At the population-level, for every kg C m$^{-2}$ increase in cNPP, $\Delta C_{veg}$ increased by 0.55 $\pm$ 0.17 kg C m$^{-2}$. While the biomass retention rate was preserved across $CO_2$ treatments within a site, the best model included a random effect on the slope suggesting that the biomass retention rate varied across sites, ranging from a non-significant 0.14 (0.07–0.55, quantiles of the standard error; Table 2, model 3) at ORNL to 0.87 (0.77–0.93) at Duke (Fig. 2; Table 1, model 3).

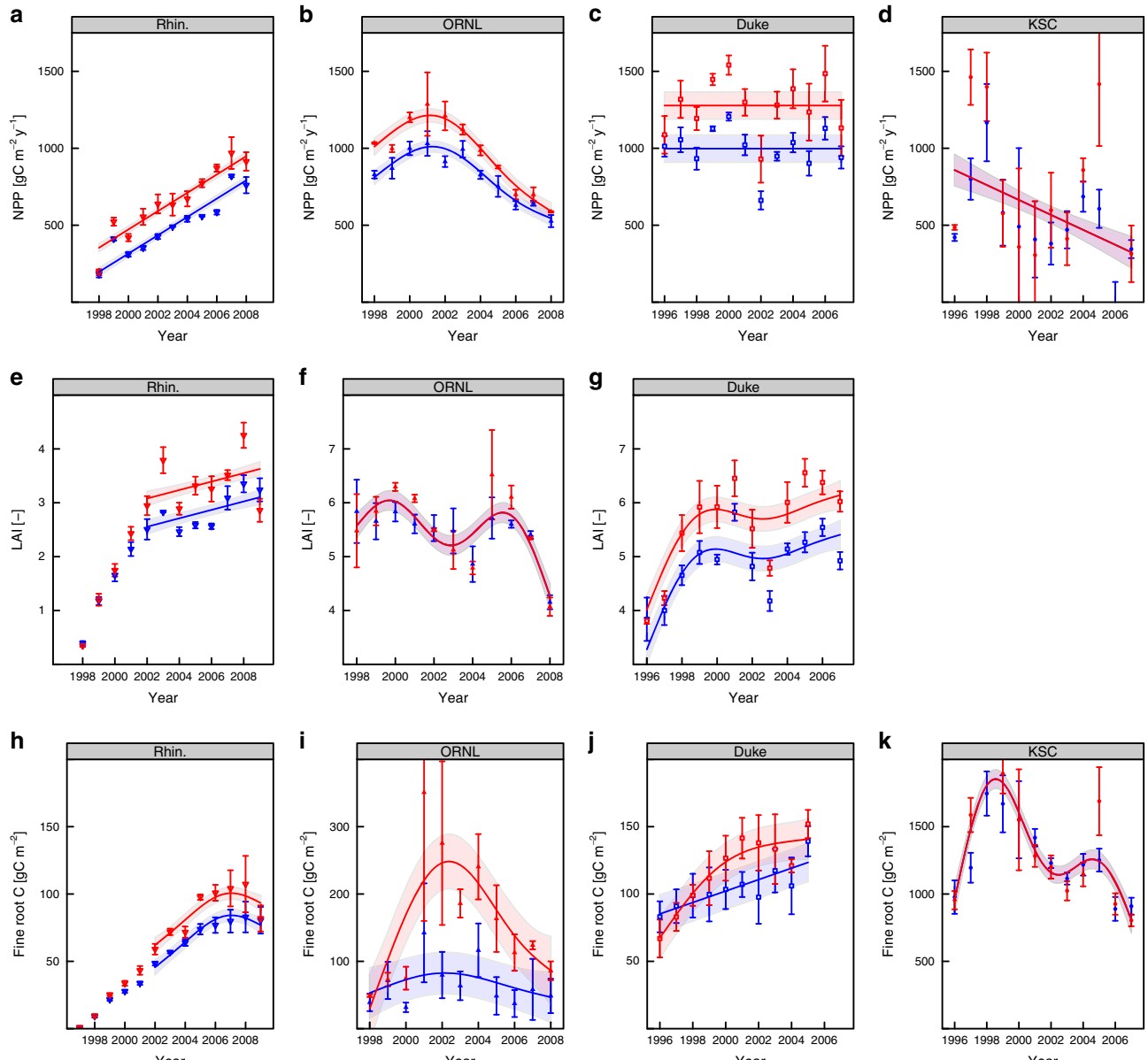

**Fig. 1** Trends in ecosystem variables to indicate successional stage. Annual net primary production (NPP; **a–d**), peak leaf area index (LAI; **e–g**) and fine-root biomass (**h–k**) dynamics over the duration of the experiments. Data show treatment (ambient shown in blue and elevated in red) means ± SEM (standard error of the mean) in each year, lines and shaded areas show the best generalised additive mixed model (GAMM) or linear models selected using corrected Akaike Information Criterion (AICc) from a set of candidate models. The number of knots in the GAMMs were determined using half the number of years in the data either minus one for even numbers of years or rounded down to the nearest integer for odd numbers (constrained to a minimum of four knots). This knot specification was intended for multi-annual trend detection that avoided over-sensitivity to inter-annual variation

Given that the $\Delta C_{veg}$ response to $CO_2$ depends upon the $CO_2$-independent biomass retention rate, we now turn to investigate the biomass retention rate in more depth. The processes governing vegetation turnover determine rates of biomass loss and therefore the period over which NPP is retained as live biomass. Vegetation turnover is determined by allocation of NPP among tissues with differing turnover rates (i.e. leaves and fine-roots vs wood) and, over longer timescales, tree mortality. To explain the variability in the biomass retention rate in these aggrading forests, where tree mortality was a relatively small fraction of total vegetation turnover, a first-order hypothesis for the fraction of cNPP remaining as $\Delta C_{veg}$ is simply the fraction of cNPP that is allocated to long-lived woody tissue (fW).

At the three sites with a biomass response to $CO_2$ enrichment, the negative intercept of the $\Delta C_{veg} \sim$ cNPP relationship (Fig. 2)

indicates that the fraction of cNPP retained as $\Delta C_{veg}$ ($\Delta C_{veg}/$cNPP; here defined as the biomass retention ratio) increased as cNPP increased. Assuming that fW is the main driver of the biomass retention ratio, constant fW across the range of cNPP would predict: 1) that the biomass retention ratio would not change with cNPP, 2) an intercept of zero in the $\Delta C_{veg} \sim$ cNPP relationship, and therefore 3) the biomass retention ratio would equal the biomass retention rate. None of these three predictions were observed, suggesting that either fW did not explain variability in biomass retention or that fW changed with cNPP. Supporting the hypothesis that wood allocation explained biomass retention, a positive linear relationship of fW with cNPP was observed (Table 2, model 3), consistent with other empirical observations[26]. The population-level increase in fW per unit cNPP was $0.02 \pm 0.005$ m$^2$ kg C$^{-1}$ and the best model did not

**Table 2 Best mixed-effects models**

| Model | Response | Fixed effect | Parameter | SEM | Random effects | | |
|---|---|---|---|---|---|---|---|
| | | | | | Re.site | Re.Intercept | Re.slope |
| 1 | $\Delta C_{veg}$ | Intercept | 3.616 | 1.156 | Rhin. | 3.320 (2.995–3.652) | – |
| | | $eCO_2$ | 1.045 | 0.258 | ORNL | 4.047 (3.698–4.376) | – |
| | | | | | Duke | 6.294 (5.913–6.585) | – |
| | | | | | KSC | 0.801 (0.825–0.614) | – |
| 2 | NPP | Intercept | 0.723 | 0.133 | Rhin. | 0.516 (0.481–0.556) | – |
| | | $eCO_2$ | 0.164 | 0.031 | ORNL | 0.814 (0.773–0.849) | – |
| | | | | | Duke | 1.050 (1.003–1.086) | – |
| | | | | | KSC | 0.511 (0.486–0.540) | – |
| 3 | $\Delta C_{veg}$ | Intercept | −0.332 | 1.422 | Rhin. | −0.245 (−1.055–0.627) | 0.642† (0.504–0.764) |
| | | cNPP | 0.546† | 0.173 | ORNL | 3.205 (−0.436–3.849) | 0.144† (0.070–0.553) |
| | | | | | Duke | −2.103 (−2.704–−0.985) | 0.873† (0.767–0.933) |
| | | | | | KSC | −2.183 (−2.640–−1.720) | 0.526† (0.460–0.594) |
| 4 | fW | Intercept | 0.365 | 0.121 | Rhin. | 0.476 (0.435–0.507) | – |
| | | cNPP | 0.020 | 0.005 | Duke | 0.480 (0.417–0.529) | – |
| | | | | | KSC | 0.139 (0.101–0.172) | – |

Model 1: mean annual NPP (kgC m$^{-2}$ y$^{-1}$) against $CO_2$ treatment; Model 2: forest biomass increment ($\Delta C_{veg}$; kgC m$^{-2}$) against $CO_2$ treatment; Model 3: forest biomass increment ($\Delta C_{veg}$; kgC m$^{-2}$) against cumulative NPP (cNPP; kgC m$^{-2}$) and $CO_2$ treatment; and Model 4: fraction of cNPP allocated to wood. Parameter values are absolute for intercept and cNPP, while the $eCO_2$ parameter is expressed as a difference from the intercept (i.e. ambient $CO_2$ parameter). $CO_2$ treatment does not appear in model 3 or 4 as it did not feature in the best models (Supplementary Tables 2 and 3). re. Intercept and re.slope show the random effect estimates of the intercept and slope for each site. Numbers in parentheses represent quantiles equivalent to the SEM of the normal distribution taken from non-parametric distributions of the random effects generated by bootstrapping model fitting with the best models.
†Indicates the biomass retention rate, i.e. the slope of the assumed linear relationship between $\Delta C_{veg}$ and cNPP.

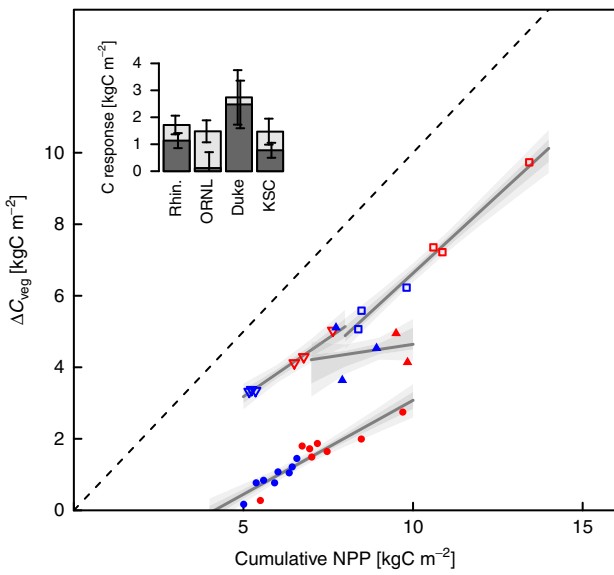

**Fig. 2** The relationship between forest biomass increment ($\Delta C_{veg}$; kgC m$^{-2}$) and cumulative NPP (cNPP; kgC m$^{-2}$) over the duration of the experiments. Each point represents an individual sample plot. Ambient plots shown in blue and elevated in red; open triangles, Rhinelander; filled triangles ORNL; open squares, Duke; & filled circles KSC. Note the generally large within-site, within-treatment variability in cNPP. Dark grey lines represent the regression from the best mixed-model (Table 2, Model 3), grey polygons represent the site-specific SEM confidence interval (CI), and the lighter grey polygons represent the 95% CI. Inset: $CO_2$ stimulation of cumulative NPP ± SEM (light grey bars) and forest biomass increment ± SEM (dark grey bars) over the duration of the experiments

include a random effect on slope (Table 3, model 3 and Supplementary Table 4), suggesting no difference among sites. As with $\Delta C_{veg}$, $CO_2$ enrichment affected fW only indirectly by increasing cNPP (Table 2, model 3 and Supplementary Table 4).

It is possible that factors other than fW (i.e. tree mortality and litterfall) were substantial contributors to vegetation turnover

and therefore the biomass retention rate. To test whether the biomass retention rate could be explained solely by wood allocation and its response to increasing production, the biomass retention rate ($d\Delta C_{veg}/dcNPP$) can be expressed as a function of the observed linear relationship of fW to cNPP (see Methods for derivation):

$$\frac{d\Delta C_{veg}}{dcNPP} = fW_a + 2fW_b c\bar{N}PP, \tag{1}$$

where $fW_a$ and $fW_b$ are the intercept and the slope of the linear relationship between fW and cNPP, and $c\bar{N}PP$ is the cross-treatment mean cNPP. This hypothesis suggests that the $\Delta C_{veg}$~cNPP relationship is quadratic, which could obviate the unrealistic negative intercept of the imposed linear relationship between $\Delta C_{veg}$ and cNPP (Fig. 1, Table 2).

At three sites with a biomass response to $CO_2$ enrichment, Rhinelander, Duke, and KSC, the calculated biomass retention rate (Eq. 1) was within the 95% CI of the observed retention rate (Table 3), indicating the key role of wood allocation and its response to $CO_2$. At Duke, the calculated biomass retention rate was very close to that observed (0.89 versus 0.87) indicating that wood allocation was likely the sole driver of the biomass retention rate. At Rhinelander, the observed biomass retention rate was lower than calculated (0.64 versus 0.73) indicating that a process other than allocation was likely increasing vegetation turnover rates (lower biomass retention rate indicates higher vegetation turnover rates). This process may have been self-thinning through stand development in the youngest forest of this study. At KSC, the observed biomass retention rate was higher than calculated (0.53 versus 0.40) indicating that vegetation turnover was decreased by a process other than a change in wood allocation. Also, estimates of the wood allocation relationship to cNPP at KSC was less precise due to higher variability in wood allocation when compared with Duke and Rhinelander (Fig. 3).

Overall the increase in wood allocation, an indirect effect of $CO_2$ via the enhancement of NPP by $CO_2$ enrichment, lowered the vegetation turnover rate thereby increasing the biomass retention ratio under $CO_2$ enrichment by a small amount from 65 to 72% (calculating Eq. 1 with population level estimates of the fW~cNPP relationship and cNPP under ambient and elevated $CO_2$).

| Table 3 Comparison of biomass retention rate (d$\Delta C_{veg}$/dcNPP) calculations | | | | |
|---|---|---|---|---|
| Site | Model 3 biomass retention rate | Biomass retention rate calculated from Eq. 1 | $fW_a$ | $2fW_bcNPP$ |
| Rhin. | 0.642 (0.39–0.89) | 0.729 | 0.476 | 0.253 |
| Duke | 0.873 (0.70–1.01) | 0.889 | 0.480 | 0.409 |
| KSC | 0.526 (0.39–0.67) | 0.404 | 0.139 | 0.265 |

These are calculated empirically (Fig. 2c and Table 2, Model 3) and calculated according to Eq. 1 from the wood allocation fraction relationship with cNPP (Table 1, model 4). $fW_a$ and $2fW_bcNPP$ represent the two terms in Eq. 1 which sum to give d$\Delta C_{veg,w}$/dcNPP. The 95% CIs are presented in parentheses for the empirical biomass retention rate

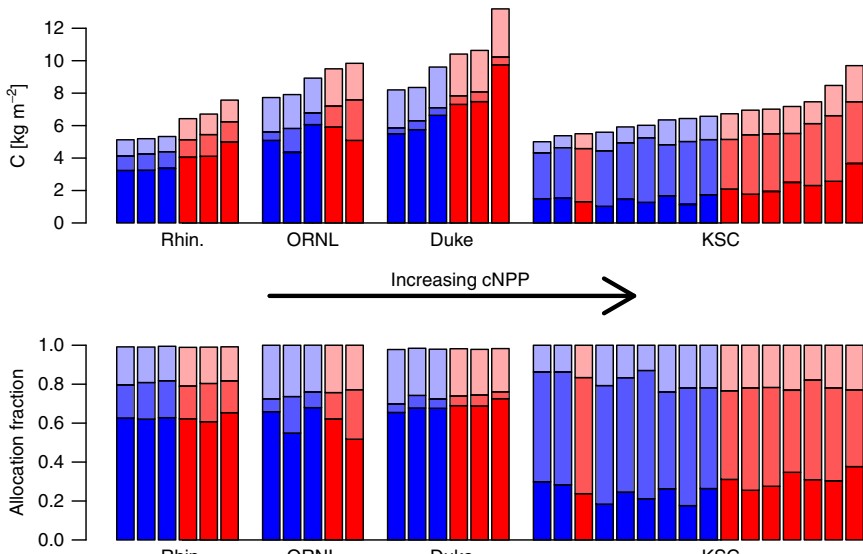

**Fig. 3** Allocation in absolute terms and fractions in each sample plot. **a** Allocation in absolute terms, **b** allocation in fractions. Ambient $CO_2$ plots in blue, elevated in red; darkest shades, wood allocation; medium shades, root allocation; and lightest shades, leaf allocation. Within each site, plots are arranged from left to right in order of ascending cumulative NPP

At ORNL, there was little evidence for a $\Delta C_{veg} \sim cNPP$ relationship due to greater uncertainty than at the other sites (Supplementary Fig. 1). The mean absolute residual from the relationship shown in Fig. 2 was 2.5 times higher at ORNL than for the site with the next highest mean absolute residual. Also, the standard error of the biomass retention rate was more than twice as large as the site with the next highest error (Table 2). This uncertainty at ORNL was caused by large within-treatment variation (0.52–0.68) in the fraction of NPP allocated to wood (fW; Fig. 3), which led to differential retention of NPP as wood.

**Ecosystem model predictions.** We used 12 state-of-the-art terrestrial ecosystem models (CABLE, CLM4.0, CLM4.5, DAY-CENT, GDAY, ISAM, JULES, LPJ-GUESS, O-CN, ORCHIDEE, SDGVM, and TECO)[27–32] to simulate these four $CO_2$ enrichment experiments and identify any common areas where model predictions of biomass carbon increment might be improved. Despite large inter-model variability, general features of the ensemble predictions are clear in relation to observations (Fig. 4). At Rhinelander and Duke, the $\Delta C_{veg}$ response to $CO_2$ enrichment was strongly under-predicted by the model ensemble (Fig. 4a). All models except one at each site predicted the $\Delta C_{veg}$ response to $CO_2$ enrichment below one standard error from the mean (Supplementary Fig. 2). At ORNL and KSC the $\Delta C_{veg}$ response was generally over-predicted by the model ensemble (Fig. 4a), though the ensemble over-prediction was less pronounced (indicated by the larger overlap between the ensemble distribution and the observed uncertainty range) than the under-prediction at Rhinelander and Duke.

Partitioning the modelled $\Delta C_{veg}$ response to $CO_2$ enrichment into the cNPP response to $CO_2$ and the $CO_2$-independent biomass retention rate, as described for the observations, allows the identification of the processes that were responsible for the model ensemble bias. At Rhinelander and Duke where $\Delta C_{veg}$ was under-predicted, the partitioning indicates that both the cNPP response to $CO_2$ and the biomass retention rate were under-predicted by the ensemble, with a stronger bias in the biomass retention rate prediction than the cNPP prediction. At ORNL, the partitioning indicates that over-prediction of the biomass retention rate was primarily responsible for the $\Delta C_{veg}$ over-prediction; whereas at KSC over-prediction of cNPP was primarily responsible for the $\Delta C_{veg}$ over-prediction. At KSC, models predicted the highest cNPP and cumulative gross primary production responses to $CO_2$ at KSC, the site with highest MAT (Table 1), most likely due to the greater C3 photosynthesis response to $CO_2$ at higher temperatures[33]. Over-prediction of the cNPP response at KSC was driven by the high predicted GPP response, supported by the largest predicted increase in N use efficiency (Supplementary Fig. 3).

At Rhinelander, ORNL, and Duke, biases in the simulation of the biomass retention rate translated to biases in the prediction of the $\Delta C_{veg}$ response to $CO_2$. Partitioning the biomass retention rate according to Eq. 1 (into the two empirical parameters $fW_a$ and $fW_b$ and the variable cNPP) shows which of these three terms were responsible for model variability and bias in the biomass retention rate (Fig. 4d–f).

Ensemble predictions of the intercept of the fW relationship with cNPP ($fW_a$) showed very large variability at each site that was much larger than the observed uncertainty (Fig. 4d).

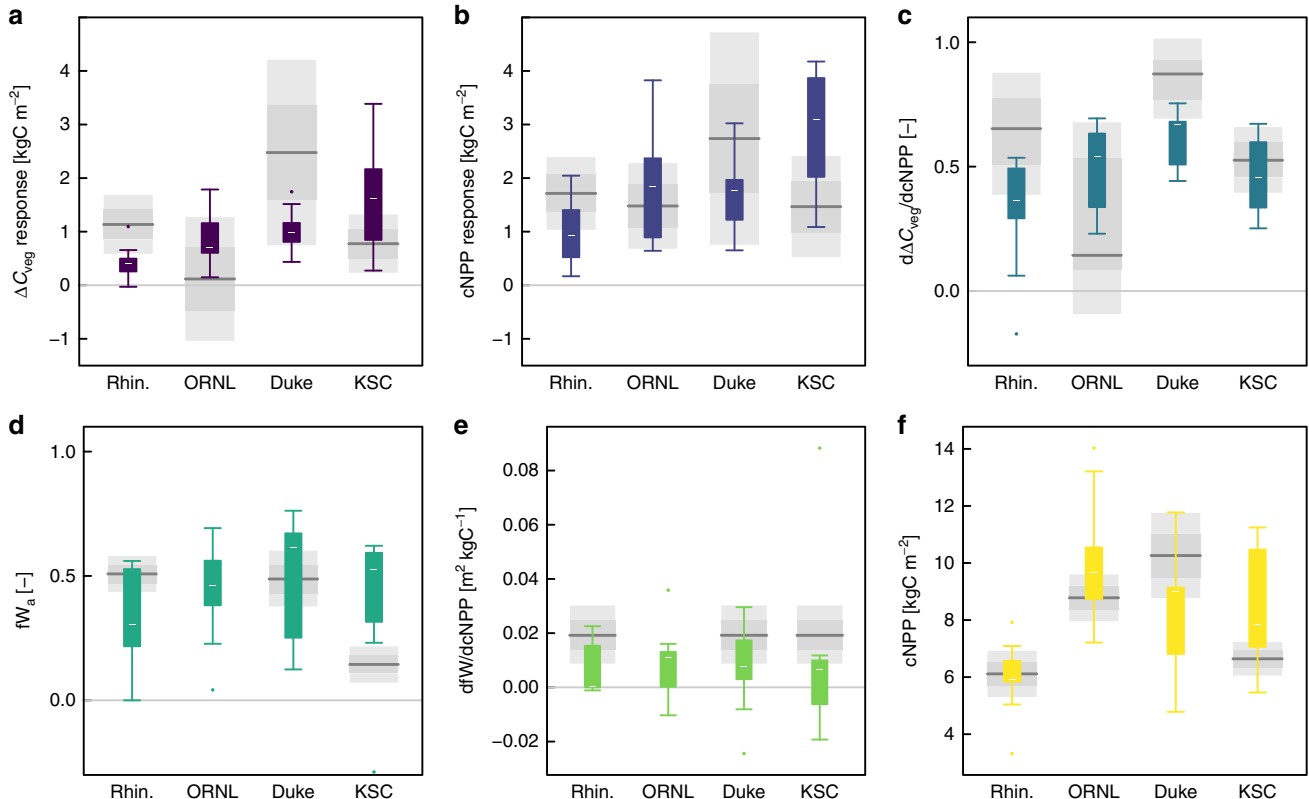

**Fig. 4** Model ensemble predictions compared against observations. **a** the $\Delta C_{veg}$ response to $CO_2$ enrichment. The key variables leading to the response— **b** the cNPP response to $CO_2$ enrichment and **c** $d\Delta C_{veg}/dcNPP$. And the three components of $d\Delta C_{veg}/dcNPP$—**d** $fW_a$, **e** $dfW/dcNPP$ and **f** cross-treatment mean cNPP. Coloured boxes and whiskers represent the model ensemble predictions (white bar is the median, the box the inter-quartile range, whiskers data within four times the IQR, and dots are outliers). Grey shaded areas represent the observations (dark grey lines are observed means or regression parameters, grey polygons are the SEM CI and the lighter grey polygons are the 95% CI). ORNL has no observed data on plots **d** and **e** as these were calculated from the regression of fW on cNPP. ORNL was not included as the regression was intended to explain the biomass retention rate, which was not significant at ORNL and was highly uncertain

However, a clear directional bias in $fW_a$ was apparent only at KSC (Fig. 4d), where the model ensemble strongly over-predicted $fW_a$. None of the models predicted $fW_a$ within the observed 95% CI at KSC. KSC had a relatively low $fW_a$, consistent with the *Quercus* species that dominate the KSC ecosystem which allocates a large fraction of growth below-ground to fine-roots at the expense of fW (Fig. 4). This allocation pattern is interpreted as an adaptation to frequent hurricane and fire disturbance that allows rapid resprouting and recovery post-disturbance[16]. The models' allocation schemes do not include such adaptation to frequent disturbance, which may account for the large over-prediction of $fW_a$ by the model ensemble (Fig. 4d).

## Discussion

The Rhinelander, Oak Ridge, Duke, and Kennedy Space Center experiments represent the most direct evidence for decadal biomass responses to $CO_2$ enrichment in early-secondary-succession, temperate woody ecosystems. The analysis shows that $CO_2$ enrichment to $CO_2$ concentrations predicted for the mid-to-late century stimulates an increase in the decadal vegetation carbon increment ($\Delta C_{veg}$) of $1.05 \pm 0.26$ kg C m$^{-2}$. This evidence suggests that at $CO_2$ concentrations of the late 1990's and early 2000's, $CO_2$ is a resource that limits decadal-scale biomass increment in early-secondary-succession, temperate woody ecosystems. The analysis also shows that the $\Delta C_{veg}$ response to $CO_2$ enrichment in these ecosystems can be predicted with knowledge of the $CO_2$ response of NPP ($0.16 \pm 0.03$ kg C m$^{-2}$ y$^{-1}$) and the $CO_2$-

independent biomass retention rate ($d\Delta C_{veg}/dcNPP$; $0.55 \pm 0.17$ at the population level), which can be calculated under *ambient* $CO_2$ conditions (Fig. 2, Table 2). This finding emphasises the importance of understanding the drivers of ecosystem variability and dynamics under current conditions in order to interpret and predict ecosystem responses to experimentally manipulated elevated $CO_2$ concentrations.

The $CO_2$ stimulation of annual NPP observed in this study is consistent with the $23 \pm 2\%$ median increase previously calculated over 1–6 years at Rhinelander, POPFACE, ORNL, Duke[34]. Mixed-model analysis allows estimation of the response of the statistical population and is the reason the uncertainty in this study is larger, 6% versus 2% in ref. [34]. The approximately linear cross-treatment relationship between $\Delta C_{veg}$ and cNPP over a decade (Fig. 2) is consistent with a previous finding for Rhinelander[35]. We have now shown this cross-treatment conservation of the biomass retention rate at all four sites, suggesting that $CO_2$ stimulated gains in NPP are retained as biomass at the same rate as variation in NPP caused by other factors. Nitrogen (N) has been shown to control much of the within treatment variability in NPP at these sites[15,25] and it is remarkable that $CO_2$-stimulated gains in NPP were retained as biomass at the same rate as NPP driven by variability in N availability. The implication is that in these ecosystems and at the decadal scale, allocation patterns were not directly affected by $CO_2$ enrichment.

The relative increase in $\Delta C_{veg}$ in response to $CO_2$ enrichment was higher than the relative increase in cNPP ($29.1 \pm 11.7\%$ versus $22.9 \pm 6.1\%$) due to a linear increase in the wood allocation

fraction (fW) as cNPP increased (Table 2; an indirect effect of $eCO_2$). Increasing fW with NPP is consistent with a comprehensive analysis of forest carbon allocation across gradients of productivity[26], and our analysis suggests that the relationship between fW and production is no different when production is affected by $CO_2$ or other factors. A simple hypothesis for the linear relationship between fW and cNPP (Table 2) is that allocation to resource acquisition organs (i.e. leaves and fine roots) at the decadal scale is fairly well conserved in these even aged stands and that variability in NPP is primarily driven by variability in wood production.

Random effects in the best statistical model of $\Delta C_{veg}$ suggest site-level differences in the biomass retention rate. Rhinelander, Duke, and KSC showed a positive biomass retention rate and thus a decadal biomass response. These three sites were in the reorganising and aggrading phases of secondary succession, with various degrees of expanding above-and-below-ground resource acquisition volumes and thus coupling to resource availability. At these sites, the biomass retention rate was tied via Eq. 1 to cNPP and wood allocation. Both the baseline wood allocation fraction $(fW_a)$ and the unit change in fW for a unit change in cNPP $(fW_b;$ which was conserved across sites) were important (Table 3). Rhinelander and Duke shared a similar $fW_a$ and the difference in cNPP at the sites determined the different biomass retention rates via the second term in Eq. 1 (Table 3). $fW_a$ was lower at KSC, leading to a lower biomass retention rate.

At ORNL, the biomass retention rate of 0.144 (−0.093–0.678, 95% CI) was not statistically different from zero and therefore there was no relationship between $\Delta C_{veg}$ and cNPP. The absence of a relationship was not a response to $CO_2$ enrichment. $\Delta C_{veg}$ at ORNL was simply more variable within $CO_2$ treatments than across treatments, related to high within-treatment variability in fW (Fig. 3) that resulted in a highly uncertain biomass retention rate. In several years of the ORNL experiment, annual root production was stimulated by $CO_2$ enrichment[36]. However, at the decadal scale, a treatment effect on allocation was not detected (Supplementary Tables 5 and 6). At ORNL, the peak and later decline in NPP in *both treatments* (Fig. 1) was attributed to progressive nitrogen limitation, which was intensified by $CO_2$ enrichment[13,24]. That both treatments were under-going PNL at ORNL suggests that stand development was the under-lying cause of the PNL[77]. In this state of tightening resource availability $\Delta C_{veg}$ was controlled by within-treatment variability in fW that was unrelated to variability in cNPP. We propose the hypothesis that cross-plot variability in the timing and intensity of competition as the plots at ORNL shifted from the aggrading phase into the transition phase of secondary succession was the cause of the within-treatment variability in fW. As with the sites in the earlier stages of succession, understanding the ambient condition and successional status is important for interpreting the (lack of) $\Delta C_{veg}$ response to $CO_2$ at ORNL.

Furthermore, the time since disturbance at Duke is greater than at ORNL, while results suggest that ORNL is in a later stage of succession. The disconnection of time since disturbance and successional stage suggests that time since disturbance, or age of the trees, may not be sufficient to indicate successional stage, which appears to be an important factor in the $CO_2$-response of biomass.

At the population level and at all four of these reorganising, aggrading, and approaching transition ecosystems, there was no evidence to suggest that the biomass retention rate was affected by $CO_2$ enrichment. The biomass retention rate was one of two variables needed to predict the $\Delta C_{veg}$ response to $CO_2$ enrichment and was calculable with knowledge of allocation and NPP under ambient conditions.

Model predictions of the $\Delta C_{veg}$ response to $CO_2$ enrichment were highly variable (Figs 4a and S2), as is common[5,37].

Nevertheless, site-specific biases across the whole ensemble were observed, indicating areas for improvement that are general to the group of models in the ensemble. At Rhinelander and Duke, the strong under-prediction of the $\Delta C_{veg}$ response to $CO_2$ enrichment resulted from under-prediction of both the cNPP response and the response of fW to cNPP. At ORNL, the over-prediction (but within observed uncertainty) of the $\Delta C_{veg}$ response resulted from over-prediction of the (highly uncertain) biomass retention rate. At KSC, the over-prediction of the $\Delta C_{veg}$ response was primarily due to over-prediction of the cNPP response, albeit that accurate predictions of the biomass retention rate were a result of compensating errors in fW prediction.

We highlight four findings related to C allocation that will help models to improve simulated $\Delta C_{veg}$ responses to $CO_2$ enrichment: (1) across $CO_2$ treatments, fW was a linear function of cNPP; (2) large variability in the predicted intercept of the relationship $(fW_a)$ led to large variability in the predicted biomass retention rate; (3) model predictions of the wood allocation response to cNPP $(fW_b)$ were low biased; and (4) models did not capture the low $fW_a$ at KSC that is assumed an adaptation to frequent disturbance. Models vary substantially in how C allocation is implemented resulting in substantial model C sink variability[32,37]. Overall, our results suggest that allocation rules were more constrained across sites than across models, though successional stage and disturbance regime did drive cross-site differences in allocation. Models with allometric constraints, such as the pipe model[38], tended to perform better in a previous analysis[32] and a representation of tree size and potentially forest structure through succession may help models to better implement the more conservative allometric constraints implied by the observations and previous analyses[26,39].

In these four ecosystems, the N constraints on NPP responses to elevated $CO_2$ were met by increased N uptake, rather than an increase in N use efficiency[31,40]. In the models with an N cycle, under-prediction of the cNPP response at Rhinelander and Duke was likely a result of overly strong N constraints that did not allow flexibility in the coupling of the C and N cycles[31,40,41]. Understanding the coupling of the C and N cycles through plant-microbe C and N dynamics and the C cost associated with N uptake will help to improve model simulations and is an ongoing area of research[31,42–46]. Furthermore, the strong nutrient constraint at ORNL, imposed by stand development, and within-treatment variability in allocation patterns makes a case for representing succession tied to dynamics of resource limitation in models.

The role of N in determining variability in NPP in these early successional ecosystems is expected[17,47]. In addition to N, other nutrients play a substantial role in areas of high N deposition, later-successional stages, and in areas with highly weathered soils (such as the tropics)[47–49]. More general understanding the effect of elevated $CO_2$ on temperate forest biomass requires knowledge of biomass responses in ecosystems across all stages of secondary succession. In a later-successional Eucalyptus woodland on very low P soils, P addition stimulated above-ground woody biomass increment over a 3-year period, while $CO_2$ enrichment did not[50]. At Flakaliden, Sweden, biomass increment in individual, 27-year-old Norway Spruce trees was increased by $CO_2$ enrichment only when a nutrient solution optimal to the species was also added[51]. In two later-successional temperate forests in Switzerland, $CO_2$ enrichment did not stimulate radial tree growth despite a stimulation of photosynthesis[8,52]. These results suggest that in these later successional forests, perhaps in the transition phase, aboveground tree biomass increment was not carbon limited. Tree-ring analyses have also found no consistent effect of the historical $CO_2$ trend on basal area increment[53,54]. Interestingly, and potentially linking our results

and those from later successional stage experiments, Voelker[20] demonstrated a decline with age in the radial growth stimulation by $CO_2$ in temperate oak and pine species. The potential for age-associated diminishing $CO_2$ responses again emphasises the need to understand the dynamics of resource limitation through secondary succession and the influence of increased photosynthate on these dynamics.

Understanding the interaction of mortality with higher rates of decadal biomass accumulation early in succession is also necessary for predicting the response of the long-term carbon sink to increasing $CO_2$[5,32,37,55]. On the premise that accelerated growth causes shorter tree longevity (i.e. higher turnover rates), it has been argued that increased growth rates caused by elevated $CO_2$ during early phases of secondary succession may not stimulate woody ecosystem biomass in the long term[11,19]. We agree that in plantation forests, where mortality (i.e., harvest) is an economic decision, higher turnover rates are a likely consequence of higher growth rates[19]. However, while there is some evidence to support the premise of increased conspecific mortality for individuals with higher growth rates[56,57], there is also a substantial body of evidence that does not support the premise, including both species-specific or site-specific studies[58–61] and an extensive multi-site synthesis[62]. The interactions of growth rate and mortality may also be important at the stand scale. Self-thinning of forest stands could be accelerated or intensified by $CO_2$-stimulated individual growth rates. Growth rates of non-dominant individuals at ORNL[63] and biomass of under-story trees at Duke[64] were both lower, but not significantly, under $CO_2$ enrichment. If increased growth rates do indeed lead to increased mortality, the immediate consequence will be increased inputs of C to the soil. How soil C responds to $CO_2$ enrichment is an active area of research that must also be considered in analyses of feedbacks between the atmosphere and terrestrial ecosystems[5,37].

The data presented here clearly show that a decade of $CO_2$ enrichment in temperate, early-secondary-succession, woody ecosystems increased vegetation carbon increment ($\Delta C_{veg}$) by about 30%. Gap dynamics are ubiquitous in primary forests, while 60% of temperate forests are naturally regenerating, secondary forests and 22% are plantation forests[65]. And the single-decade scale of these FACE and OTC studies is the temporal scale at which the carbon cycle becomes relevant to climate change. Thus post-disturbance stands and early successional forests are likely to be a major component of the climate-relevant, temperate forest responses to increasing $CO_2$. Nevertheless, four sites is a small sample size of the temperate woody ecosystem population, a single decade is at the lowest end of the decadal scale, and a full range of secondary succession was not sampled. Secondary succession, gap dynamics, and the dynamics of limiting resources through successional stages, provides a context for scaling $CO_2$ responses to greater spatial and temporal scales. To implement secondary succession as a context for accurately scaling predictions of terrestrial ecosystem biomass responses to increasing $CO_2$ requires further development and synthesis of mechanistic theory. In particular, we need to understand how stand development across successional stages influences: wood allocation in relation to NPP, the interaction of $CO_2$ with other resources to limit plant production, and the interaction of NPP, mortality, and self-thinning.

## Methods

**Experiments**. The Rhinelander FACE experiment was established on moderately fertile sandy loam soils at the Harshaw Experimental Farm of the USDA Forest Service, Wisconsin (45.6 °N, 89.5 °W) following 20 years of plantation forestry, pre-dated by 50 years of agricultural use. Small trees (~25 cm tall) were planted in 1997 at 1 m spacing in one of three community types: aspen (*Populus tremuloides* Michx.), equal parts aspen and birch (*Betula papyrifera* Marshall), or equal parts aspen and maple (*Acer saccharum* Marshall). Only the mixed-genotype aspen

community was used in this analysis. Climate is the fully humid, warm-summer, continental cold climate of the Köppen-Geiger classification[66] (mean annual temperature 4.9 °C, mean annual precipitation 800 mm). The other three sites' climates are classified as fully-humid, hot-summer, warm temperate[66].

The ORNL FACE experiment was located in a sweetgum (*Liquidambar styraciflua* L.) plantation on the Oak Ridge National Environmental Research Park, Tennessee (35.90 °N, 84.33 °W). The forest is on a low fertility silty-clay loam and the climate at the site is typical of the humid southern Appalachian region (mean annual temperature 13.9 °C and mean annual precipitation 1370 mm). The trees were planted in 1988 and at the start of the experiment in 1998 the trees had a fully developed canopy.

The Duke FACE experiment was located within a 90 ha loblolly pine (*Pinus taeda* L.—Piedmont provenance) plantation situated in the Duke Forest, Chapel Hill, North Carolina (35.97 °N, 79.08 °W). The forest is on a moderately low fertility acidic loam and the climate is typical of the warm-humid Piedmont region of the south-eastern US (mean annual temperature 15.5 °C and mean annual precipitation 1150 mm, with precipitation evenly distributed throughout the year). The trees were planted in 1983 and the experiment was initiated in late 1996.

The Kennedy Space Center experiment occurred at the Merritt Island National Wildlife refuge, Florida (28.63 °N, 80.70 °W). After controlled burning, open-top chambers were established over the regrowing scrub oak (*Quercus spp.*) vegetation. The experiment began in May 1996. The soils at the site are sandy with a low pH (c. 4) classified as Arenic Haplahumods and Spodic Quartzipsamments and the climate is subtropical (mean annual temperature 22.1 °C and mean annual precipitation 1094 mm).

Our selection criteria excluded three woody ecosystem $CO_2$ enrichment experiments: POPFACE, WebFACE, and EucFACE from the analysis, as they were either managed, did not quantify NPP and biomass, or had been running for just 5 years, respectively. We have a maximum of 11 years of data for each experiment. Experiment data used in this study are freely available[67].

All components of NPP and biomass were measured, though somewhat different methods were used at each site. For our analyses, allocation fractions were calculated as the organ production divided by total NPP (which was calculated as the sum of production of all organs: leaves, wood, coarse-roots, fine-roots). Details of how the measurements were made can be found in Table 4 and the references cited therein. At three of the sites methods were consistent across years while at Rhinelander different methods in the early and later parts of the experiment were carefully combined[35]. Ecosystem live-biomass calculations were some combination of mean (commonly fine-roots), peak (leaves and fine roots at some sites), and point-in-time measurements (wood at the end of the growing season). Across sites, woody biomass was calculated from measured diameter (and other non-destructive measurements in some cases) in conjunction with measured site and species-specific allometric models (Table 4). Annual net primary production (NPP) was calculated as the sum of annual dry matter production of all plant organs using a combination of the above method for biomass to calculate wood increment, canopy assessments and litter traps, mini-rhizotrons and root coring (Table 4). Dry matter units were converted into carbon units using organ-specific carbon concentrations.

For the allocation analysis coarse roots were combined with stem wood for a total 'wood' pool. Biomass and NPP components at Duke are the sum of both evergreen over-storey and hardwood under-storey. Biomass and NPP at Rhinelander are from only the tree components of the ecosystem and do not include the herbaceous component (which made a minimal contribution).

An issue in all meta-analysis type studies is that variability in the way that measurements have been made may result in artifacts that obscure true differences across experiments. Random effects in mixed models account for unidentified variability across samples within a group when calculating population level fixed-effects, but cannot account for unidentified variability when specific sample-to-sample comparisons are made. The decadal biomass increment is mostly a result of wood increment, which was measured using site and species-specific allometric functions and detailed measurements across sites, as described above. Thus the primary result of a $1.05 \pm 0.26\ kg\ C\ m^{-2}$ stimulation of biomass increment by $CO_2$ enrichment is likely robust. Measurements within each experiment site were consistent, which suggests that our second major result, that the biomass retention rate is independent of $CO_2$, is also likely robust. Measurements of fine-root production are likely to vary the most among sites, though each method was state-of-the-art, and are thus most likely to influence cross-site comparisons. For example, at KSC fine-root biomass was quantified with a combination of cores and mini-rhizotrons, and coarse roots with a combination of allometry and ground-penetrating radar. Although the same suite of methods was not used at other sites, we have no reason to believe such differences explain 5–10 times higher root biomass at the KSC site. Furthermore, at two sites (ORNL and KSC) with different methods of measuring root production, and largely different values of fine-root production, the measurements of production both under ambient $CO_2$ and in response to $CO_2$ enrichment were supported by measurements of fine-root biomass during final harvests[68,69]. Thus while the cross-site comparisons of the exact value of the biomass retention rate may be coloured by methodological differences, we expect these biases to be small.

KSC used biomass increment and litterfall to calculate NPP[69]. In a few years this calculation resulted in negative NPP in some plots because biomass (in particular, fine-root biomass; Fig. 1l) fluctuated strongly year-to-year but calculations of litterfall did not fluctuate as strongly. Based on the conceptualisation

**Table 4 Comparison of various methods used to calculate biomass and NPP at the four sites**

| | Site | | | |
| --- | --- | --- | --- | --- |
| | **Rhinelander** | **ORNL** | **Duke** | **KSC** |
| **Biomass** | | | | |
| Leaves | 2002–2008, littertraps. Pre-2002, allometric relationship. | Littertraps. | Littertraps, lagged for pines. | Diameter based allometric functions. |
| Wood | Diameter based allometric functions. Two functions were used depending on a diameter based cutoff. | Diameter and height based functions relationships, annual measurements of wood carbon density. | Diameter and height based allometric functions, annual measurements of wood density. Sub-sample of full plot. | Diameter based allometric functions. |
| Coarse-root | Linear function of above-ground tree mass and fine-root mass. | Diameter based allometric functions. | Function of above-ground biomass. | Soil cores, ground-penetrating radar, and allometric functions (when cores and GPR were not taken). |
| Fine-root | 2002–2008, mini-rhizontrons. Pre-2002, allometric relationship. | Mini-rhizotrons. | Soil cores. | Mini-rhizotrons and soil cores. |
| **Production** | | | | |
| Leaves | Equal to biomass. | Equal to biomass. | Peak Leaf Area Index divided by species specific SLA. | Biomass increment plus litterfall. Litterfall estimated from littertraps. |
| Wood | Biomass increment at the tree scale*. | Biomass increment at the tree scale*. | Plot scale biomass increment. | Biomass increment plus litterfall. Litterfall assumed zero. |
| Coarse-root | " | " | " | Same method as fine roots. |
| Fine-root | 2002–2008, in-growth cores and mini-rhizontrons. Pre-2002, biomass increment plus estimated root litterfall from 2002–2008 mini-rhizotron data. | Mini-rhizotrons. | Biomass multiplied by proportion of annual length production from mini-rhizotrons. | Biomass increment plus litterfall. Litterfall estimated as biomass multiplied by C turnover rate. Turnover rate measured using an isotopic tracer approach. |
| for details see | Talhelm et al.[35] | Norby et al.[13] | McCarthy et al.[15] | Hungate et al.[16,69] |

*Accounts for mortality such that mortality is not included in this estimate and the minimum NPP for this variable is zero

of NPP used at the other three sites (i.e. the sum of gross production), negative values of NPP at KSC mean that components of either litterfall or biomass were not measured. Negative NPP and not knowing whether the unmeasured litterfall or biomass component of NPP varies annually or not makes annual time-scale comparisons with the other three sites difficult. At multi-annual timescales the large fluctuations in biomass that cannot be accounted for with litterfall measurements are averaged leaving a single directional bias, a bias that is likely present and similar in both cumulative NPP and biomass.

**Statistical analysis**. We used mixed-model analyses with site as the random effect because the data were unbalanced and the particular sites and ecosystems in which the experiments were conducted represent a 'population' (in the statistical rather than the biological sense) of ecosystems (temperate, early-secondary-successional, woody ecosystems) for which we are interested in $CO_2$ responses. Mixed-effects models are capable of handling unbalanced data and have generalisable conclusions due to their assumption that the groups within the random effects are random samples drawn from a population[70]. Although the sites in our analysis were not strictly selected at random, they span a range of climates, soil types and ecosystems. We analysed all sites together in a unified framework using mixed-effects models, treating sites as random effects in a way that is analogous to blocks in a randomised block design experiment. The advantage of using mixed-effects models is their ability to analyse data gathered across multiple individuals within a single statistical model and to handle unbalanced data. In treating individuals—in this case sites—as samples drawn from a population distribution, mixed-effects models allow the determination of a population level fixed effect including the uncertainty in that effect. The fixed effects provide population level estimates of the effect of $CO_2$ on NPP and biomass increment, i.e. an estimate of the general effect of $CO_2$ in semi-natural aggrading temperate forests. Random effects allow an estimation of the variability in fixed effects among groups and thus whether there are differences across sites.

Given the evolving landscape of mixed-effects modelling methods, for model fitting we used the 'lmer' function within the up-to-date 'lme4' R package[71]. All models were fit using maximum likelihood (ML) estimation of the parameters to allow comparison of models with different fixed-effects using Akaike Information Criterion corrected for finite sample size (AICc)[72]. Once the fixed effects terms were selected according to the minimum adequate model, models with the same fixed effects but different random effects were fit using restricted maximum likelihood (REML) parameter estimates as REML gives a more robust estimate of the random effects[70]. We were interested in the site-level parameter estimates so accurate fitting of the random effects was desired. Confidence intervals for the site-level parameter estimates in the minimum adequate model were generated by

bootstrapping using the 'bootMer' function in 'lme4', i.e. the minimum adequate model was refit 1000 times by resampling the data.

From a full model that included all main effects and interactions, model simplification was exhaustive (all possible combinations of main effects and interactions were tested, subject to the inclusion of a main effect if the variable was also considered in interaction). A null model that included only the random effect on the intercept was included in the model selection to ensure that models with fixed effects contained information in addition to simply the sites having different mean values. Model selection, based on Akaike Information Criterion corrected for finite sample size (AICc), was used to find the 'minimum adequate model' where the model with the lowest AICc was considered the minimum adequate model. The AIC is a metric of fit to the data while also considering parsimony in the number of parameters used in the model, i.e. the minimum adequate model can be thought of as the model that simultaneously maximises the fit to the data and parsimony[70,72]. For brevity in the main text we refer to the minimum adequate model as the best model, specifically this means the model that minimises the Kullback–Leibler distance indicating that minimum information is lost in the model[72].

The 'gamm4' R package was used to calculate the generalised additive mixed models (GAMMs) used in the trend detection. The GAMMs were fit to each variable (NPP, LAI, and fine-root biomass) at each site individually. Random effects were the treatment ring, fixed effects were year and $CO_2$ treatment. Model selection was conducted as described above. Confidence intervals were generated using the 'predict' function.

All statistical analyses were conducted in R v3.3.2[73] using the 'lmer' function for mixed effects modelling[71] and the 'lm' function from the 'stats' package for fixed-effects modelling. AICc's were calculated using the 'AICcmodavg' package in R[74]. Unless otherwise stated error bars describe one standard error of the mean (SEM) with an $n$ of four at Duke, eight at Kennedy Space Center, three and two in the ambient and elevated treatments at ORNL, and three at Rhinelander.

**Analysis of production and biomass increment relationship**. The mixed-model analysis provides a rigorous statistical approach to analyse results from all the sites together in a single statistical model. However, the resulting statistical model is empirical and does not immediately provide information on mechanism. Below we derive a set of equations designed to mechanistically interpret the empirical parameters of the linear mixed-effects models. Assuming a linear empirical relationship to describe $\Delta C_{veg}$ as a function of NPP over multiple years (i.e. the statistical model that is fit to the data):

$$\Delta C_{veg,e-s} = a + b \sum_{t=s}^{e} NPP_t, \qquad (2)$$

where $a$ and $b$ are the empirical intercept and slope of a linear relationship, the subscripts $s$ and $e$ are the time at the start and the end of the experiment, and $\sum_{t=s}^{e} \text{NPP}_t$ is cumulative NPP over the time period in question (cNPP as referred to in the main text). Equation 2 is equivalent to the more process oriented:

$$\Delta C_{\text{veg},e-s} = \sum_{t=s}^{e} \text{NPP}_t - \sum_{t=s}^{e} L_t, \qquad (3)$$

where $L_t$ is litterfall in year $t$. The advantage of starting our analysis with Eq. 2 is that it links directly to the mixed-effects statistical analysis. To describe the response of $\Delta C_{\text{veg}}$ to $CO_2$ enrichment Eq. 2 can be used to represent the response of $\Delta C_{\text{veg}}$ from ambient $CO_2$ (subscript amb) and elevated $CO_2$ (subscript ele) treatments:

$$\begin{aligned} \Delta C_{\text{veg},e-s,\text{response}} &= \Delta C_{\text{veg},e-s,\text{ele}} - \Delta C_{\text{veg},e-s,\text{amb}} \\ &= (a_{\text{ele}} + b_{\text{ele}}\text{cNPP}_{\text{ele}}) - (a_{\text{amb}} + b_{\text{amb}}\text{cNPP}_{\text{amb}}), \end{aligned} \qquad (4)$$

and assuming the slope and the intercept of Eq. 3 do not respond to $CO_2$ enrichment, simplified to:

$$\Delta C_{\text{veg},e-s,\text{response}} = b(\text{cNPP}_{\text{ele}} - \text{cNPP}_{\text{amb}}) = \frac{d\Delta C_{\text{veg}}}{d\text{cNPP}}(\text{cNPP}_{\text{ele}} - \text{cNPP}_{\text{amb}}), \qquad (5)$$

giving the response of $\Delta C_{\text{veg}}$ to $CO_2$ enrichment as a function of the cNPP response to enrichment and $b$. As the differential of Eq. 2, $b$ is the rate of change in $\Delta C_{\text{veg}}$ with respect to cNPP ($d\Delta C_{\text{veg}}/d\text{cNPP}$).

Given that $b$ is an empirical parameter, an analysis to explain $b$ in terms of biological processes is now proposed, which also allows the evaluation of the models in terms of process. Hypothesising that the primary cause of the relationship between $\Delta C_{\text{veg}}$ and cNPP is wood allocation, such that:

$$\Delta C_{\text{veg},e-s} = fW\text{cNPP}, \qquad (6)$$

where $fW$ is the fraction of cNPP allocated to wood. Again, to link with the statistical analysis, assuming that wood allocation follows a linear function of cNPP:

$$fW = fW_a + fW_b\text{cNPP}, \qquad (7)$$

were $fW_a$ and $fW_b$ are the empirical intercept and slope of a linear relationship. Substituting Eq. M6 into Eq. M5 gives the quadratic:

$$\Delta C_{\text{veg},e-s} = (fW_a + fW_b\text{cNPP})\text{cNPP}. \qquad (8)$$

A quantitatively testable hypothesis that the biomass retention rate estimated from the empirical relationship of Eq. 2 is controlled by wood allocation is that at the mean cNPP the differential of Eq. 2 (i.e. $b$) is equal to the differential of Eq. 8:

$$\frac{d\Delta C_{\text{veg}}}{d\text{cNPP}} = fW_a + 2fW_b\text{cNPP}. \qquad (9)$$

**Models**. Twelve terrestrial biosphere/ecosystem/carbon cycle models (TBMs) were used to simulate the four experiments. The models were applied to the sites following a common protocol which specified meteorological data, $CO_2$ data, common parameterisations of soil characteristics, plant traits and site land use history[75,76]. Meteorological data went through rigorous quality control and standardisation. The protocol and data can be found on the FACE model data synthesis webpage (facedata.ornl.gov/facemds). The models we used were seven global land surface models: CABLE, CLM4.0, CLM4.5, ISAM, JULES, O-CN, and ORCHIDEE; two global dynamic vegetation models: LPJ-GUESS and SDGVM; and three ecosystem models DAYCENT, GDAY and TECO (see 26–31 for model descriptions). Nine of these models simulate a process-based mass-balanced N cycle while JULES, ORCHIDEE and SDGVM simulate only the carbon cycle (SDGVM considers an empirical N limitation on photosynthetic rates).

The modelling protocol specified site histories to ensure these simulations were in a successional stage similar to the ecosystems of the experiments. Models that were in equilibrium at the beginning of the simulation of the experiments were excluded from the analysis. The modelling protocol required two simulations—one ambient $CO_2$ and one elevated $CO_2$—at each site, meaning the determination of model relationships of $fW$ with cNPP using regression was not possible. As a surrogate for the linear slope we use the difference in $fW$ divided by the difference in cNPP between the simulated treatments, and plot $dfW/d\text{cNPP}$ against the wood allocation fraction in the simulated elevated $CO_2$ treatment (Fig. 4c, f, i, l). For the TBMs we used mean standing crop within a year to calculate biomass. Meteorological data, model output, and protocols are freely available[75,76].

## Data availability
The site-based meteorological dataset (https://data.ess-dive.lbl.gov/view/ess-dive-7807cf86f1dd42a-20181127T173047368940), the model output dataset (https://data.ess-dive.lbl.gov/view/ess-dive-8260043c35fc925-20181130T171955541030)

and the experiment dataset (https://data.ess-dive.lbl.gov/view/ess-dive-f525c71da7d2681-20181128T160851574946) generated and analysed during the current study are available at the US Department of Energy's (DOE) ESS-DIVE repository.

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

## Acknowledgements

The FACE Model-Data Synthesis project was supported by the US Department of Energy, Office of Science Biological and Environmental Research program. Oak Ridge National Laboratory is operated by UT-Battelle, LLC, under contract DE-AC05-00OR22725 with the US Department of Energy. M.D.K. also acknowledges support from the ARC Centre of Excellence for Climate Extremes (CE170100023). Support for J.P.M. and M.L. was provided by a DOE-TES program grant DE-SC0008339 and the Smith-sonian Institution. A.K.J. and S.S. were supported by the US National Science Foundation (NSF-AGS-12-43071). R.O. was supported by the Erkko Visiting Professor Programme of the Jane and Aatos Erkko 375th Anniversary Fund through the University of Helsinki.

## Author contributions

A.P.W. wrote the paper and conceived of and conducted the analysis. M.G.D.K., B.E.M., S.Z., C.M.I. and R.J.N. also wrote the paper. E.R. contributed to statistical design. All authors contributed to writing the paper. A.P.W., M.G.D.K., S.A., B.G., A.H., T.H., A.K.J., Y.L., X.L., K.L., S.S., Y.P.W., C.W. and J.X. ran the models. R.J.N., R.O., D.R.Z., A.T., B.A. H., C.M.I., P.M., J.M.W., M.L., A.P.W., B.Y., and many more ran the experiments, collected experimental data and collated the experimental data.

## Additional information

**Competing interests:** The authors declare no competing interests.

Anthony P. Walker[1], Martin G. De Kauwe[2], Belinda E. Medlyn[3], Sönke Zaehle[4], Colleen M. Iversen[1], Shinichi Asao[5], Bertrand Guenet[6], Anna Harper[7], Thomas Hickler[8,9], Bruce A. Hungate[10], Atul K. Jain[11], Yiqi Luo[10], Xingjie Lu[12], Meng Lu[13,14], Kristina Luus[15], J. Patrick Megonigal[14], Ram Oren[16,17], Edmund Ryan[18], Shijie Shu[11], Alan Talhelm[19], Ying-Ping Wang[12], Jeffrey M. Warren[1], Christian Werner[8], Jianyang Xia[20,21], Bai Yang[1], Donald R. Zak[22] & Richard J. Norby[1]

[1]Environmental Sciences Division and Climate Change Science Institute, Oak Ridge National Laboratory, Oak Ridge 37831-6301 TN, USA. [2]ARC Centre of Excellence for Climate Extremes, University of New South Wales, Sydney 2052 NSW, Australia. [3]Hawkesbury Institute for the Environment, Western Sydney University, Locked Bag 1797, Penrith, NSW 2751, Australia. [4]Biogeochemical Integration Department, Max Planck Institute for Biogeochemistry, Hans-Knöll-Str. 10, 07745 Jena, Germany. [5]Natural Resource Ecology Laboratory, Colorado State University, Fort Collins, CO 80523-1499, USA. [6]Laboratoire des Sciences du Climat et de l'Environnement, LSCE/IPSL, CEA-CNRS-UVSQ, Université Paris-Saclay, F-91191 Gif-sur-Yvette, France. [7]College of Engineering, Mathematics, and Physical Sciences, Laver Building, University of Exeter, Exeter EX4 4QF, UK. [8]Senckenberg Biodiversity and Climate Research Centre (BiK-F), Senckenberganlage 25, 60325 Frankfurt/Main, Germany. [9]Department of Physical Geography, Geosciences, Goethe-University, Altenhöferallee 1, 60438 Frankfurt, Germany. [10]Center for Ecosystem Science and Society, Northern Arizona University, Flagstaff, AZ 86011, USA. [11]Department of Atmospheric Sciences, University of Illinois, 105 South Gregory Street, Urbana, IL 61801-3070, USA. [12]CSIRO Oceans and Atmosphere, Private Bag #1, Aspendale, Victoria 3195, Australia. [13]School of Ecology and Environmental Science, Yunnan University, Kunming 650091, China. [14]Smithsonian Environmental Research Center, Edgewater, MD 21037, USA. [15]Centre for Applied Data Analytics Research (CeADAR), Dublin Institute of Technology, Camden Row, Dublin 4, Ireland. [16]Nicholas School of the Environment & Pratt School of Engineering, Duke University, Durham, NC 27708, USA. [17]Department of Forest Sciences, University of Helsinki, FI-00014 Helsinki, Finland. [18]School of Mathematics, University of Manchester, Manchester M13 9PL, UK. [19]Department of Forest, Rangeland, and Fire Sciences, University of Idaho, Moscow, ID 83844, USA. [20]Tiantong National Station of Forest Ecosystem and Research, Center for Global Change and Ecological Forecasting, School of Ecological and Environmental Sciences, East China Normal University, Shanghai 200241, China. [21]Institute of Eco-Chongming (IEC), 3663 N. Zhongshan Rd., Shanghai 200062, China. [22]School of Natural Resources and Environment, and the Department of Ecology and Evolutionary Biology, University of Michigan, Ann Arbor 48109 MI, USA

