## [Peer Review File · Nature Communications]

Reviewers' comments:

Reviewer #1 (Remarks to the Author):

Comment on Walker et al. Nature Communications

The authors provide a statistical analysis of the outcome of 4 different US CO₂-enrichment experiments with various woody test systems in order to advance the theory of potential CO₂ fertilization of forests. They then applied 12 different ecosystem models to compare experimental results with model predictions. While stats and modeling reflect state of the art, the way the analysis was designed and is communicated is misleading for the wide audience this paper aims for.

(1) The term forest. To a wide audience this suggests something very different to what these test systems actually represent. A shrub oak bush is by no means a forest, nor are assemblages of seedlings that were allowed to grow into thickets in an experimental farm. The two young, mono-specific plantations at Duke and Oak Ridge are the only systems that come close to what one could still address as a forest. I do not think the study would be devaluated scientifically if the authors acknowledged the developmental stage of the experimental stands and used the term 'forest' more cautiously. In fact, I was surprised that successional stage wasn't used as a predictor. Why only mention that Oak Ridge trees were in a linear growth phase? I assume that the Rhinelander FACE was in an early exponential phase for most of the experiment's duration, certainly in the early years. The Duke one was possibly at the transition from exponential to linear, and the KSC site had been burned, so the early years covered an 'explosive' phase of regeneration, given the extreme imbalance between below ground and above ground biomass. As long as the models do not account for such differences in developmental stage, it is hardly surprising that they underestimate the responses of the two extremely young systems and overestimate the response of the plantation in Oak Ridge that arrived at steady state growth. In contrast, the Duke site had not reached steady state LAI by the end of the experiment (Kim et al. (2016) Glob Change Biol 22: 944-956).

(2) When biomass accumulation is expressed as the net increment since the beginning of the experiment (line 125), it makes a difference if starting biomass was close to zero (KSC) or tiny (Rhinelander), or already substantial such as at Duke and Oak Ridge. It seems this could have contributed to the effect size in stats.

(3) If 'aggrading forest' is defined as 'accumulating biomass' (line 65) this refers to almost ALL temperate forests, unless we talk about very old growth forest, ready to collapse. So 'aggrading' refers to a triviality, namely that forests grow until they are harvested, burnt or wiped out by a hurricane. Aggrading forests are not only a substantial component (390-392) of, but represent almost all forests in the US outside Alaska. What is a non-aggrading forest? one that does not accumulate biomass (line 383)? Even old growth tropical forests are aggrading biomass if one subtracts the 1-5 % of forest area that is in a gap phase. The 60 % temperate forests that are suggested to be self-regenerating, are accumulating biomass as well, except for the small fraction under collapse, harvest, or in a natural gap phase. The entire § 383-396 is misleading, because it confuses the degree of 'aggrading' with the developmental stage of individuals/populations, the issue of the current paper. The more important aspect is that plant production had been examined along a gradient from near to bare ground after fire to closed forest tree canopy.

(4) While meteorological data underwent rigorous quality control and standardization (line 455), the far more important questions are not touched upon: (1) did soils (growth conditions, nutrients) undergo the same quality check and standardization? (2) Were test systems checked/standardized for the developmental status, or LAI during the early phase of CO₂ enrichment? (3) were the soil moisture implications of CO₂ enrichment controlled/standardized?

(5) Replication: For CO₂, replication goes from 2 (Oak Ridge) to 8 plots (KSC) per site. So the KSC data have more weight than the Oak Ridge data in the overall model? What if KSC researchers had decided to have 16 smaller instead of 8 bigger OTCs? So my point is that the replication must be weighted by the size of the system tested. Subdividing a test field in smaller units may create less variance for that reason only. The Oak Ridge data are weighing less because of less replication (greater variance; line 157-162), although this is the most realistic test system of the four, because the canopy had been closed before CO₂ enrichment started. From Fig. 2 one gets the impression that the 'Duke effect' comes from the third ring. Any idea why that ring responded differently? Without that ring, the data look quite similar to Oak Ridge. If someone had decided to use an 80 year old hard wood forest near Duke campus, parcel replication might not have been feasible for the sheer size of those trees. But assume such trees had been studied (which come closer to what people consider forest trees), the replication per unit ground area would be zero. In fact, later on, I noticed that the authors dismissed the value of such data in the literature for that reason, rather than capitalizing on the full catena, from resprouts after disturbance and nursery to fully grown trees in a natural forest. This is the dilemma of the entire research field, that the more realistic the test conditions become, the less feasible the test becomes, hence we end up with replicated systems that are far away from representing our forests in the US or elsewhere in the world. Hence, my dismay, when seeing the outcome of the analysis being sold as 'forest responses' (e.g. line 256, 266, 347).

(6) Finally I do not think the authors are doing justice to the matter in question by how they are embedding their analysis in a C sink debate to add political weight. This starts with a problematic statement in the abstract that suggests that NPP may automatically lead to a C sink, and ends with the conclusion in line 403 that these results demonstrate that such a sink is likely to exist. Mitigating atmospheric CO₂ requires growing terrestrial pools and whether these pools will rise depends on tree turnover which can only be identified by repeated inventories in permanent plots that include all age classes. Fast rotation systems store less. See the details below.

(7) Summary. For this to become a representative analysis of how forests (and their trees) respond to elevated CO₂, across all developmental stages, the latter (and all its attributes) must be made the corner stone of the analysis. The authors would be well advised to handle the available scientific evidence 'inclusively' rather than excluding works for arbitrary reasons. This attributes the paper a defensive mood. The current results do not reflect 'best knowledge' (line 347), but reflect the bias in the state of tree and population development within the 4 test systems. Only one study included here arrived at steady state of LAI and litter production, statistically drowning among the signals that emerge from the 'by chance' inclusion of early successional systems. The 'ecosystem scale' is an unsuitable and unneeded discriminant, given that any population level response in a forest could be scaled to unit forest area for pool size estimates. It is trees that build the biomass, irrespective of spatial setting. By disregarding soil carbon exchange (I agree on that) the authors already made a decision in favor of tree populations, rather than ecosystem C balance (no NEP or NEE data).

Details on nutrient limitation and the NPP-tree longevity debate

(8) Nutrient limitation

Line 354: I do not think the situation is that special. For old Gondwana land surfaces this is the normal situation, with the dominant taxa well adapted to that situation. For tropical forests, N is not a rate limiting factor (still half of the globe's forests). Many temperate forests of the northern hemisphere are P limited and/or are running into a critical decline of the P/N ratio. N-limitation is an issue of young and disturbed systems (the type of systems studied here except for the shrub oaks during the later years of the experiment) and not as universal in the temperate zone as

presented here and as models might assume. See for instance:

Vitousek PM (1984) Litterfall, nutrient cycling, and nutrient limitation in tropical forests. *Ecology* 65:285-298

Wardle DA, Walker LR, Bardgett RD (2004) Ecosystem properties and forest decline in contrasting long-term chronosequences. *Science* 305:509-513

Li Y, Niu SL, Yu GR (2016) Aggravated phosphorus limitation on biomass production under increasing nitrogen loading: a meta-analysis. *Glob Change Biol* 22:934-943

Braun S, Thomas VFD, Quiring R, Fluckiger W (2010) Does nitrogen deposition increase forest production? The role of phosphorus. *Environ Pollut* 158:2043-2052

(9) Tree vigor, tree longevity and tree height... this issue is too central to be bypassed by 'we are unaware'.... (367)

Line 367-369: In the forest industry, the answer is obvious: A faster growing tree arrives at harvest size faster. For unmanaged forests see for instance:

Bugmann H, Bigler C (2011) Will the CO₂ fertilization effect in forests be offset by reduced tree longevity? *Oecologia* 165:533-544

For Colorado conifers:

Bigler C, Veblen TT (2009) Increased early growth rates decrease longevity of conifers in subalpine forests. *Oikos* 118:1130-1138

For the tropics:

Brienen RJW et al. (2015) Long-term decline of the Amazon carbon sink. *Nature* 519:344-348

Also in a species comparison, fast growing taxa such as poplar, aspen, but also birch and maple are shorter lived than slower growing taxa such as oaks or beech, the combination of fast growth and long life does not seem to exist in tree species, perhaps due to the tradeoff described by Loehle, or because there is a maximum species-specific tree height, or at community level, because of the distinction between r and K strategists. When r strategist tree taxa would profit from CO₂ enrichment and suppress K strategists, the carbon stock would decline.

Loehle C (1998) Height growth rate tradeoffs determine northern and southern range limits for trees. *J Biogeogr* 25:735-742

The following paper is a rather trivial statistical exercise, but may contain references for the vigor/age/height tradeoff (I did not check):

Stahl U, Reu B, Wirth C (2014) Predicting species' range limits from functional traits for the tree flora of North America. *Proc Natl Acad Sci U S A* 111:13739-13744

The following review covers major US forest tree species:

Purves DW (1989) The demography of range boundaries versus range cores in eastern US tree species. *Proc R Soc B* 276:1477-1484

Reviewer #2 (Remarks to the Author):

The paper provides an interesting and frankly very critical, thorough and smart analysis of decadal forest biomass response to atmospheric CO₂. The Authors analyze four sites across a large climate gradients that have been exposed to ~10 years of enriched CO₂. The analysis integrates these four sites with appropriate statistical modeling, while tying in simple concepts of biomass accumulation, but also confronts forest models and/or land surface model. The major findings are that forest biomass gains more under higher level of CO₂. Analysis of biomass gains show relationship with net primary productivity and wood allocation. Models are not capable of capturing the breadth of responses across sites, due to the lack of being able to correctly reproduce one or several factors that contribute to the CO₂ response (NPP, NPP response or the response of vegetation increase in response to NPP).

While a very nice and thorough analysis, a few questions remain.

Novelty of the CO₂ response finding: The FACE experiments have been running for a long time and quite a large body of literature exists detailing biospheric response to atmospheric CO₂, including NPP response, growth responses etc. It is not clear, how this study relates to those earlier findings. For example, to what degree is the number of 22.6 % different than the NPP increase found in Norby et al., 2006? Yes, you translated the NPP response into a biomass increase, and yes you showed that wood allocation increases with NPP? What does that mean to the interpretation of earlier results?

It is not super clear to me why the biomass increase is linear to cumulative NPP when wood allocation is a first order response to it? Wouldn't your finding of a positive fw_b lead to a (albeit small) quadratic increase?

It is not clear to me why the authors choose to include a lengthy discussion of soil carbon response and aggrading vs. non-aggrading systems. While this offers some context, this discussion brings in often information that is less relevant to the paper. I suggest considerable shortening starting at line 372. Similarly, the last sentence of the abstract also seems to distract from the major finding and can be shortened. These additions seem as the authors want to make sure that there is more to the story of CO₂ and climate response.

I like very much the inclusion of the models, and particularly the decomposition of the response into several factors, these are critical pointers to move models (that should be capable to capture a CO₂ response) forward.

Detailed Comment:

Abstract: Population-level results. This becomes clear when reading the method section of the paper. But from the abstract it almost reads like the response is evaluate based on the population of trees. I suggest revising this.

L63: Total Biomass increment increases linearly with cNPP, does this not contradict your finding of a positive fw_b, and the fact that the linear relationship has a negative intercept? Please explain in the main text what exactly is meant.

L68: second half of the sentence starting with albeit: This is not a finding of this study, I suggest to delete. I understand that there is a need for caveats, but this is probably better placed in the conclusion.

Line 75: Suggest to write: "This uncertainty arises PARTLY due...", there are other uncertainties

L 86: typo "experiment THAT lasted"

L133: typo: explanatory

L285: First sentence: This has not been tested, but assumed in the model (I do not oppose that assumption). However, it reads like as this has been statistically tested.

L322: Not clear whether "predicted" belongs to the statistical model devised here, or the mechanistic models.

L327: Wood allocation is a linear function of cNPP. This is not clear. Is it the fraction allocated to wood that is a linear function of NPP (Equation 2), or the amount of NPP that is allocated to wood is constant (absolute vs. relative terms)? This is a bit confusing and I hope this can be a bit better explained (also together with the statement elsewhere that Biomass increment increases linearly with cNPP.

L372: I appreciate the authors acknowledging that this is not whole system C accumulation, but I think this caveat can be considerably shortened since this was not really part of the system.

L383: Also this paragraph has the potential to be shortened (and perhaps put in a single paragraph combining all the caveats and limitation of the study.

L398: Again, another paragraph for shortening potential.

Reviewer #3 (Remarks to the Author):

The purpose of this study is to determine if net primary productivity is limited by carbon and if the biomass increment is significant at decadal time scales in aggrading temperate forests (line 96-97). Methods included both a statistical approach, and an approach using ecosystem models. Ecosystem models were also examined to identify any common areas where the models could be improved (line 213). [As Kaume et al 2014 New Phytologist p885 says 'we did not statistically evaluate models against observations because models can easily yield quantitatively good responses for incorrect reasons; thus such an approach typically does not correctly diagnose model deficiencies'] Both of these approaches relied on four experimental sites that involved CO₂ enrichment for a period over ten years.

The authors have used a complex statistical approach to tease information out of the experimental data, and much of the text focuses on the more complex items. It is the lack of information about the basics of the study, or the way the information is presented that makes it difficult to follow the manuscript.

Clarification about the basics of the study, or including the basics, is needed. In the title and several places in the text, the study is about aggrading temperate forests, and yet table 1 clearly lists the KSC site as being 'mature'. Lines 348-356 discuss "the only CO₂ enrichment experiment replicated at the forest stand scale in a mature forest", and the need to understand responses in non-aggrading forests (with the implication that non-aggrading forests are mature – which is what the word mature usually implies: no longer aggrading). Why is KSC included in a study on aggrading forests if the site is mature and therefore possibly not aggrading? Also, in the manuscript KSC is referred to as sub-tropical. How is sub-tropical related to temperate? Reasoning through this, I wonder why KSC is included in this study on aggrading temperate forests because it appears at first look to not be aggrading, nor is it temperate?

The term 'semi-natural' is occasionally used in the text which gives the reader pause. What is meant by this? Of the four sites, three are plantations, and the one that is not a plantation is: KSC. Here is a third item where this site is different (not to mention it is the only site which is an OTC rather than a FACE). Why does this site belong in this study? A paragraph stating why these sites are included in this study, that these are the only 4 sites in the world that meet the criteria for inclusion in this analysis, is necessary.

Based on the initial results, however, the experimental site that seems to not fit well is ORNL. It is also the only site that does not seem to have a 0.55 biomass production rate, which appears to be more of a biomass production ratio. (for every unit of NPP, what is the resulting biomass increment?) I wonder what the results for ORNL would look like if the 0.55 was imposed on the calculations?

Biomass increment (line 125-126) was measured as the 'difference between biomass in the final year minus biomass in the initial year of the experiments'. Please clearly define succinctly what components are included in biomass increment. Going through the additional methods, it appears this includes coarse roots, and all live vegetation (includes understory but does it include foliage of all live vegetation?). It appears that a few measurements were taken, and then biomass was calculated using formulas. There may be measurements, but biomass is not measured, it is estimated or modeled. Also, by definition, any vegetation that died over the time period simply drops out, meaning if there was a lot of mortality biomass increment could be negative.

In Figure 1, in the inset in the top left corner, could mortality be the reason for the small increase in biomass increment as compared to cumulative NPP? (One would think so, by definition.) To truly understand what is being attributed to as a biomass increment response to CO₂, the manuscript needs to clearly and concisely articulate the simple information/calculations needed to come up with those estimates. What biomass equations were used to estimate biomass increment? Were these equations used to estimate NPP? Was NPP measured at each site each year, or are NPP measured/estimated across some years? In the additional methods section, I can pick up some of this information but having important factors summarized in a succinct table for easy comparison would be very useful to help the reader understand possible fundamental differences in results from each site.

If this was a study focused only on ecosystem models and processes the above information may not be as important. But these items matter in an empirical approach.

This study could be vastly improved with one close read by an author, ensuring the basics of the sites and study were well-covered and consistent throughout. Something as simple as using a spell-checker would be a nice touch for the reader, given there are random mis-spellings throughout which distract (For example, line 133 expanatory; line 158 realltionship; line 369 increaded). Line 320-321: am wondering if the term 'highly uncertain' is more accurately 'highly variable'?

The purpose of the study needs to be improved, by stating it explicit and crisp. Be clear why these four sites are all appropriate to include – were any sites omitted that could have qualified? (such as in Europe)? There is a discussion about biases that is confusing. Does the statistical technique induce bias? Is bias being used more in terms of the ecological models? Is it 'true' bias in the statistical sense, or is the term 'bias' simply being used to indicate poor model fit, perhaps due to throwing data together which a statistician would omit as outlier? Or an ecologist may omit because the site does not meet the description of sites under study?

In summary, the complex approaches used here may be solid, but improved explanations are needed, especially on the basics. The topics being discussed are quite important, but the manuscript needs to be improved.

Response to Referees: manuscript NCOMMS-17-24567

Revised title: **Decadal biomass increment in early secondary successional woody ecosystems is increased by CO₂ enrichment**

We appreciate the generally supportive and constructive nature of the reviews, and we thank the reviewers for their time. We have carefully considered each of the reviewers' comments and incorporated the majority of them in the revised manuscript which, in our view, has strongly improved the study. In responding to their comments we feel the study is more robust, better framed, and more readily interpretable.

As we describe in detail below, we have substantially reframed the study, revised and improved our definition of terms, and added more information to the methods. We conducted new analyses of the trends in NPP, LAI, and fine-root biomass to provide a quantitative placement of these sites in developmental stage. Finally, we have removed the political context, removed the discussion of soils, but maintained the discussion of the link between growth rates and turnover. Please see our detailed responses (added in blue for ease of reading, with direct quotations from the revised manuscript in bold) to the points made by each of the reviewers below. Where text has been edited in the manuscript, line numbers are provided for the main manuscript and the tracked changes manuscript (line numbers prefixed with tc).

Reviewers' comments and detailed responses:

Reviewer #1 (Remarks to the Author):

The authors provide a statistical analysis of the outcome of 4 different US CO₂-enrichment experiments with various woody test systems in order to advance the theory of potential CO₂ fertilization of forests. They then applied 12 different ecosystem models to compare experimental results with model predictions. While stats and modeling reflect state of the art, the way the analysis was designed and is communicated is misleading for the wide audience this paper aims for.

We were certainly not being deliberately misleading, but we agree that some of our use of terms and their definitions could have been more precise and could be improved. We have made a substantial effort to be more precise in our terminology, and to tie the terminology in with the existing literature. We appreciate the reviewer's concern that the studies represent a bias in terms of secondary successional stage and we agree that the context of secondary succession of these results is extremely important. We hope the reviewer agrees that we have done a much better job of framing this analysis in the context of secondary successional stage.

We also agree that perhaps the political context was somewhat unnecessary, though given the nature of the study area the results do have political implications. Nevertheless, we have removed the political context in the framing of the study and focused on biological conclusions.

(1) The term forest. To a wide audience this suggests something very different to what these test systems actually represent. A shrub oak bush is by no means a forest, nor are assemblages of seedlings that were allowed to grow into thickets in an experimental farm. The two young, mono-specific plantations at Duke and Oak Ridge are the only systems that come close to what one could still address as a forest. I do not think the study would be devaluated scientifically if the authors acknowledged the developmental stage of the experimental stands and used the term 'forest' more cautiously. In fact, I was surprised that successional

stage wasn't used as a predictor. Why only mention that Oak Ridge trees were in a linear growth phase? I assume that the Rhinelander FACE was in an early exponential phase for most of the experiment's duration, certainly in the early years. The Duke one was possibly at the transition from exponential to linear, and the KSC site had been burned, so the early years covered an 'explosive' phase of regeneration, given the extreme imbalance between below ground and above ground biomass. As long as the models do not account for such differences in developmental stage, it is hardly surprising that they underestimate the responses of the two extremely young systems and overestimate the response of the plantation in Oak Ridge that arrived at steady state growth. In contrast, the Duke site had not reached steady state LAI by the end of the experiment (Kim et al. (2016) Glob Change Biol 22:944-956).

The comment about the term “forest” is a fair point and we have rephrased our labeling of these ecosystems as: “early secondary successional woody ecosystems.” We note that the use of the term aggrading was an attempt to acknowledge the secondary successional stage of these systems though we recognise that our use of this term was unclear (see below for more detail). We agree that KSC does not meet accepted definitions of forest, and also agree that Rhinelander is not exactly what most people think of as a forest (but noting that Rhinelander has been described many times as a forest in the literature).

In the revised manuscript we present new analyses of trends in NPP, LAI, and fine-root biomass in these ecosystems that place these systems in the context of secondary successional stage. Linear mixed-models and generalized additive mixed models (GAMMs) to test for curvilinear trends both were used for trend detection. This analysis shows that NPP at KSO indeed showed “explosive” regrowth early in the experiment. The analysis also suggests that NPP at Rhinelander increased with a linear trend throughout the experiment. However, at Duke, NPP showed no such trend, in opposition to the reviewer's comment that Oak Ridge was the only site in steady-state growth. The analysis also shows that LAI at Duke, although highly variable, showed no clear trend in the latter two thirds of the experiment. We could find no evidence in Kim et al. (2016) to suggest that LAI was increasing at Duke (and confirmed by one author of Kim et al. Who is also a co-author here). We conclude that Duke was in a linear growth phase, and should be designated as “coupled” in the classification scheme of Körner (2006). The influence of nitrogen availability on NPP at Duke (McCarthy et al., 2010) also supports this conclusion. Also, during the experiment at ORNL, NPP peaked and was in decline at the end of the experiment. We interpret this as the ecosystem transitioning from the “aggrading” phase to the “transition” phase of Bormann and Likens (1979).

Text added to the introduction can be found on lns 94-96 and tc105-107:

Our selection criteria restrict the analysis to temperate ecosystems dominated by woody plants (woody ecosystems), in the early phases of secondary succession.

lns 104-105 and tc118-120:

... i.e. an estimate of the general effect of CO₂ in the population of early secondary-successional, temperate woody ecosystems.

and lns 120-125, and tc134-139:

The secondary successional stage of these ecosystems is also an important factor in interpretation of any CO₂ responses. The shifts in resource availability associated with secondary succession¹⁶⁻¹⁸ are likely key factors influencing the CO₂ response and evidence suggests that CO₂ responses reduce under tightening resource coupling associated with succession¹⁹ and with the age of individuals²⁰. Thus we also interpret our results in the successional context.

The new Figure 1 is shown below and text describing this figure can be found on lns 128-181, tc142-195.

Figure 1. Annual NPP (a-d), LAI (e-h) and fine-root biomass (i-l) dynamics over the duration of the experiments. Data show treatment (ambient shown in blue and elevated in red) means \pm SEM in each year, lines and shaded areas show the ‘best’ GAMM or linear models selected using AICc from a set of candidate models. The number of knots in the GAMMs were determined using the half number of years in the experiment either minus one for even numbers of years or rounded down to the nearest integer for odd numbers (constrained to a minimum of four knots). This knot specification was intended for multi-annual trend detection that avoided over-sensitivity to inter-annual variation.

(2) When biomass accumulation is expressed as the net increment since the beginning of the experiment (line 125), it makes a difference if starting biomass was close to zero (KSC) or tiny (Rhineland), or already substantial such as at Duke and Oak Ridge. It seems this could have contributed to the effect size in stats.

Starting biomass could have influenced the subsequent biomass accumulation and response. Therefore we added a new statistical model that included time since disturbance at the beginning of the experiment as a quantitative metric related to developmental stage (and approximating starting biomass). The statistical model that included time since disturbance and NPP as additive effects was competitive with some of the other statistical models (i.e. had lower AICc values), though it had a higher AICc than the 'best' model (delta AICc of 2.4). The effect of age since disturbance in this model was actually mildly positive on biomass increment with a coefficient of 0.11 (kg C m⁻² y⁻¹ since disturbance). We include this model in the table in the supplement, but do not discuss in the main text.

(3) If 'aggrading forest' is defined as 'accumulating biomass' (line 65) this refers to almost ALL temperate forests, unless we talk about very old growth forest, ready to collapse. So 'aggrading' refers to a triviality, namely that forests grow until they are harvested, burnt or wiped out by a hurricane. Aggrading forests are not only a substantial component (390-392) of, but represent almost all forests in the US outside Alaska. What is a non-aggrading forest? one that does not accumulate biomass (line 383)? Even old growth tropical forests are aggrading biomass if one subtracts the 1-5 % of forest area that is in a gap phase. The 60 % temperate forests that are suggested to be self-regenerating, are accumulating biomass as well, except for the small fraction under collapse, harvest, or in a natural gap phase. The entire § 383-396 is misleading, because it confuses the degree of 'aggrading' with the developmental stage of individuals/populations, the issue of the current paper. The more important aspect is that plant production had been examined along a gradient from near to bare ground after fire to closed forest tree canopy.

This is an important point and one that is most fundamental to the interpretation of these experiments in the broader context of CO₂ fertilisation of the terrestrial carbon sink. Our use of the term "aggrading" was an attempt to acknowledge the inference space of these experiments, though we recognise that this and our use of a number of other related terms was unclear. We now use the term "aggrading" much more sparingly and in the specific context of the secondary successional stages described by (Bormann and Likens 1979). We now place the developmental/successional stage of these experiments into the existing classification literature on the stages of secondary succession in forests (Bormann and Likens 1979, 1994; Körner 2006). While neither of these schemes are strictly quantitative, we interpret a quantitative analysis of the trends and dynamics of a number of variables (net primary production, NPP ; peak leaf area index, LAI; and fine-root biomass) to link these ecosystem to their developmental stage.

(4) While meteorological data underwent rigorous quality control and standardization (line 455), the far more important questions are not touched upon: (1) did soils (growth conditions, nutrients) undergo the same quality check and standardization? (2) Were test systems checked/standardized for the developmental status, or LAI during the early phase of CO₂ enrichment? (3) were the soil moisture implications of CO₂ enrichment controlled/standardized?

Meteorological data under-went strict quality control for the modelling activity. For the model runs: (1) soil texture, water holding capacity, and rooting depth were standardised and given to the modellers. (2) Site histories were reconstructed and models were run during the transient phase (1850 to the year before initiation of the experiment) according to these site histories, with some judgement left to modellers to allow reasonable LAI and NPP at the beginning of the simulations. (3) Soil moisture implications would have been simulated by the models according to their conceptualisations of these processes.

If for the analysis of the experimental results: (1) site-level differences on the fixed effects of CO₂ are accounted for by the specification of random effects. (2) In response to the above comments, we

have tried to do a better job of placing these experiments in the context of their developmental status. (3) Soil water effects are implicit in these analyses, we do not attempt to explicitly analyse the interactive effects of CO₂ and soil water in this study as this was not our focus. In part this was because many of these studies only measure soil water in the top soil layers (15-30 cm) and not throughout the rooting profile (75 – 200 cm). Different soil water effects across the sites are accounted for in the random effects of the statistical model. It was never our intention to draw inferences about soil water interactions.

(5) Replication: For CO₂, replication goes from 2 (Oak Ridge) to 8 plots (KSC) per site. So the KSC data have more weight than the Oak Ridge data in the overall model? What if KSC researchers had decided to have 16 smaller instead of 8 bigger OTCs? So my point is that the replication must be weighted by the size of the system tested. Subdividing a test field in smaller units may create less variance for that reason only. The Oak Ridge data are weighing less because of less replication (greater variance; line 157-162), although this is the most realistic test system of the four, because the canopy had been closed before CO₂ enrichment started. From Fig. 2 one gets the impression that the 'Duke effect' comes from the third ring. Any idea why that ring responded differently? Without that ring, the data look quite similar to Oak Ridge. If someone had decided to use an 80 year old hard wood forest near Duke campus, parcel replication might not have been feasible for the sheer size of those trees. But assume such trees had been studied (which come closer to what people consider forest trees), the replication per unit ground area would be zero. In fact, later on, I noticed that the authors dismissed the value of such data in the literature for that reason, rather than capitalizing on the full catena, from resprouts after disturbance and nursery to fully grown trees in a natural forest. This is the dilemma of the entire research field, that the more realistic the test conditions become, the less feasible the test becomes, hence we end up with replicated systems that are far away from representing our forests in the US or elsewhere in the world. Hence, my dismay, when seeing the outcome of the analysis being sold as 'forest responses' (e.g. line 256, 266, 347).

1) *Unbalanced replication and its influence within a mixed-model.* Mixed-models account for unbalanced data (Pinheiro and Bates, 2009), giving close to equal weight to each group. This was a key reason for choosing mixed-effects models, as described in the original manuscript. Therefore, although ORNL had only 2 replicates in the elevated CO₂ treatment and KSC had 8, the sites were equally weighted (and not weighted according to their number of replicates in the statistical model).

2) *Duke effect due to one ring.* It is well understood why biomass was much higher in the ring described by the reviewer – as part of natural variability at the site, nitrogen availability was higher in that ring (McCarthy et al. 2010). Differences in N availability at the site were known and ambient and elevated rings were paired according to N availability.

We emphatically disagree that the removal of this point alone makes the Duke data look like the data from ORNL (see Figure below, left panel). The regression lines and confidence intervals are from the model applied to the full Duke dataset. While the range of NPP and biomass is lower, a strong linear relationship is still apparent (unlike ORNL) that falls on the same line as the full dataset. A clear CO₂ effect on biomass increment is still visible (also unlike ORNL). Removing the ambient ring with higher N availability (as one should if removing the one elevated ring with higher N availability) is also shown below (right panel) and the NPP response and Cinc_{veg} response is clear.

Figure S2 (& 2) of manuscript with subset of data. Duke (open squares) and Oak Ridge (closed triangles). Left panel, removal of highest NPP plot in elevated CO_2 treatment at Duke. Right panel, removal of highest NPP plot in both CO_2 treatments at Duke. Regression lines and error polygons are from the statistical model with all of the plots at Duke included, i.e. they are the exact same as those shown in Figure 2 and S2. It is clear that the regression very well describes the remaining five or four plots at Duke. Also clear is that neither subset of the Duke data appear similar to the Oak Ridge data.

3) “... the dilemma of the entire research field, that the more realistic the test conditions become, the less feasible the test becomes, hence we end up with replicated systems that are far away from representing our forests ...” We think this is a valid statement and considering replication in terms of ground area is a good point that we had not considered. The dismissal of studies of larger individual trees in the original manuscript was a little rash and we have removed this statement and revised the associated text (lns 422-425, tc484-490) as follows:

In a later-successional *Eucalyptus* woodland on very low P soils, P addition stimulated above-ground woody biomass increment over a three year period, while CO_2 enrichment did not⁴⁶. In two later-successional temperate forests in Switzerland, CO_2 enrichment did not stimulate radial tree growth despite a stimulation of photosynthesis^{8,47}.

That being said, the experiments in our analysis are representative of temperate forests (and woody ecosystems) early in the stages of secondary succession or gap phase dynamics and even steady-state forest are composed of a mosaic of secondary successional phases. We would really like to include more studies across a broader range of successional stage in this analysis. However, the few CO_2 manipulation studies in later successional stage woody ecosystems did not quantify NPP or biomass, so are not suitable for our analysis.

(6) Finally I do not think the authors are doing justice to the matter in question by how they are embedding their analysis in a C sink debate to add political weight. This starts with a problematic statement in the abstract that suggests that NPP may automatically lead to a C sink, and ends with the conclusion in line 403 that these results demonstrate that such a sink is likely to exist. Mitigating atmospheric CO_2 requires growing terrestrial pools and whether these pools will rise depends on tree turnover which can only be identified by repeated inventories in permanent plots that include all age

classes. Fast rotation systems store less. See the details below.

We mostly agree that framing in terms of the political debate was probably a step too far. We have removed the final statement of the abstract and deleted the concluding paragraph of the manuscript.

In the originally submitted manuscript, the statement in the abstract to which the reviewer refers read:

Stimulation of photosynthesis by increasing atmospheric CO₂ often results in increased net primary production (NPP), but at longer timescales, may not result in increased biomass.

We disagree that this statement suggests that “NPP may automatically lead to a C sink.” Whether increased NPP in response to elevated CO₂ leads to increased biomass increment at longer timescales is *the primary question* of the study. We have edited the first paragraph of the introduction to be clearer that increased GPP may not lead to increased NPP, and that increased NPP may not lead to increased biomass (lns 74-82, tc83-91):

This uncertainty [in Earth System model projections] arises, in large part, due to limited predictive understanding of the ecosystem processes that determine the fate of additional photosynthate arising from the stimulation of photosynthesis by CO₂ enrichment^{6,7}. Higher atmospheric CO₂ may not stimulate forest net primary production (NPP) if the supply of photosynthate does not limit NPP at ambient CO₂ concentrations⁸, as in cases where the availability of other resources (e.g. nitrogen; N) limit NPP⁹. Even if NPP is stimulated by CO₂ enrichment, tree biomass may not increase if the additional NPP is allocated to fast-turnover plant organs¹⁰ or if tree mortality rates increase¹¹. Long-term, ecosystem-scale CO₂ enrichment experiments provide the most direct evidence of whether rising atmospheric CO₂ may lead to increased forest biomass carbon.

We also agree that tree turnover is a key piece of the puzzle and determinant of the CO₂ response of the terrestrial carbon sink, as we discuss in response to the reviewer’s point #9 below.

(7) Summary. For this to become a representative analysis of how forests (and their trees) respond to elevated CO₂, across all developmental stages, the latter (and all its attributes) must be made the corner stone of the analysis. The authors would be well advised to handle the available scientific evidence 'inclusively' rather than excluding works for arbitrary reasons. This attributes the paper a defensive mood. The current results do not reflect 'best knowledge' (line 347), but reflect the bias in the state of tree and population development within the 4 test systems. Only one study included here arrived at steady state of LAI and litter production, statistically drowning among the signals that emerge from the 'by chance' inclusion of early successional systems. The 'ecosystem scale' is an unsuitable and unneeded discriminant, given that any population level response in a forest could be scaled to unit forest area for pool size estimates. It is trees that build the biomass, irrespective of spatial setting. By disregarding soil carbon exchange (I agree on that) the authors already made a decision in favor of tree populations, rather than ecosystem C balance (no NEP or NEE data).

While we agree that an analysis of forest responses to elevated CO₂ across all developmental stages would be ideal, it was never our intention to do this, primarily because experiments across all developmental stages do not exist. Our intention was to analyse the relationship between biomass increment and NPP at the decadal timescale of a decade (the longest time period over which these experiments have been run). The only experiment run for close to a decade in a later developmental stage forest is that at Basel in Switzerland (Körner 2005, Bader 2012). But they did not quantify NPP nor biomass. A goal was also to capture whole plant responses that include root responses. Our

analysis already includes all of the experiments that have been run for a decade, in unmanaged woody ecosystems (i.e. not pots), and that have measured NPP and biomass.

As suggested, we present a quantitative placement of these ecosystems within a secondary successional context and have made this context the cornerstone of the framing of the study.

While not included in the analysis for the above stated reasons, we also have been more inclusive of the later developmental stage experiments in the discussion.

We do not agree that the Oak Ridge site was the only site in steady-state. The trend analysis shows that Duke had the most steady NPP, albeit variable across years. At ORNL the trend analysis suggests ORNL was not in steady-state: LAI in the ambient treatment was not steady-state, nor was NPP. We classify ORNL as being on the boundary of aggrading and transitional in the scheme of Bormann and likens (1979).

Details on nutrient limitation and the NPP-tree longevity debate

(8) Nutrient limitation

Line 354: I do not think the situation is that special. For old Gondwana land surfaces this is the normal situation, with the dominant taxa well adapted to that situation. For tropical forests, N is not a ratelimiting factor (still half of the globe's forests). Many temperate forests of the northern hemisphere are P limited and/or are running into a critical decline of the P/N ratio. N-limitation is an issue of young and disturbed systems (the type of systems studied here except for the shrub oaks during the later years of the experiment) and not as universal in the temperate zone as presented here and as models might assume. See for instance:

Vitousek PM (1984) Litterfall, nutrient cycling, and nutrient limitation in tropical forests. *Ecology* 65:285-298

Wardle DA, Walker LR, Bardgett RD (2004) Ecosystem properties and forest decline in contrasting long-term chronosequences. *Science* 305:509-513

Li Y, Niu SL, Yu GR (2016) Aggravated phosphorus limitation on biomass production under increasing nitrogen loading: a meta-analysis. *Glob Change Biol* 22:934-943

Braun S, Thomas VFD, Quiring R, Fluckiger W (2010) Does nitrogen deposition increase forest production? The role of phosphorus. *Environ Pollut* 158:2043-2052

We take the reviewer's point that other nutrients are likely to be important and revise the text to that effect (lns 432-443, tc501-512):

In addition to N, other nutrients play a substantial role in areas of high N deposition, later-successional stages, and in areas with highly weathered soils (such as the tropics)⁵⁰⁻⁵². Therefore, understanding the interaction of phosphorus, potassium, and other nutrients with increasing CO₂ will be necessary for predictive understanding outside of temperate, early successional ecosystems.

(9) Tree vigor, tree longevity and tree height... this issue is too central to be bypassed by 'we are unaware'.... (367)

We agree that this is a crucial component to how inferences of from experiments can be extended to forest across greater spatial and temporal scales. Hypothetically speaking, if one assumes that plant turnover is not influenced directly or indirectly by CO₂, our results of increased biomass accumulation rates imply that mean forest biomass will increase in response to CO₂ even if later developmental stage forest biomass accumulation is not stimulated by CO₂. This is because across

both space and time, mean forest biomass will be higher with higher accumulation during early stages of secondary succession. As the reviewer points out, the question then becomes whether increased rates of biomass accumulation influences tree longevity. If longevity is decreased, any increased biomass accumulation rates are likely to be canceled out, as demonstrated by the modelling study of (Bugmann and Bigler, 2011). On the other hand, if longevity is increased by higher growth rates then this will complement higher growth rates. See further comment below.

Line 367-369: In the forest industry, the answer is obvious: A faster growing tree arrives at harvest size faster.

It is a good point that increased biomass accumulation rates lead to earlier harvesting dates in forestry plantations and we acknowledge this in the manuscript (lns 449-451, tc520-521).

For unmanaged forests see for instance:

Bugmann H, Bigler C (2011) Will the CO2 fertilization effect in forests be offset by reduced tree longevity? *Oecologia* 165:533-544

For Colorado conifers:

Bigler C, Veblen TT (2009) Increased early growth rates decrease longevities of conifers in subalpine forests. *Oikos* 118:1130-1138

For the tropics:

Brienen RJW et al. (2015) Long-term decline of the Amazon carbon sink. *Nature* 519:344-348

The key question is: Do higher individual growth rates lead to shorter life-spans of those individuals in unmanaged forests. In our opinion only one of these references (Bigler & Veblen, 2009) cited by the reviewer actually shows that higher growth rates of individuals (within populations and within species) leads to shorter lifespan in that individual. Bugmann & Bigler 2011 is primarily a modelling study that *assumes* faster growth leads to higher turnover based on strategies *across tree species*, not for individuals within populations. Brienen et al. (2015) shows a declining rate of biomass accumulation caused by increasing forest mortality. But the cause of the increase in mortality was not identified.

Prompted by R1's comments we searched the literature and we note that there are many studies of the link between individual growth rates and mortality that show a positive correlation (the opposite to R1's assertion) or show no relationship, in particular Cailleret et al (2017). We augment the discussion of turnover as follows (lns 445-454, tc514-525):

Furthermore, understanding the interaction of higher decadal rates of biomass accumulation early in succession with mortality is necessary for understanding changes in the long-term carbon sink^{31,36,53}. Based on the premise that accelerated growth causes shorter tree longevity (i.e. higher turnover rates), it has been argued that increased growth rates caused by elevated CO₂ during early phases of secondary succession may not stimulate woody ecosystem biomass in the long term^{11,19}. We agree that in plantation forests, where mortality (i.e., harvest) is an economic decision, higher turnover rates are a likely consequence of higher growth rates¹⁹. However, while there is some evidence to support the premise of increased conspecific mortality for individuals with higher growth rates^{54,55}, there is a broader body of evidence that does not support the premise, including both species-specific or site-specific studies⁵⁶⁻⁵⁹ and extensive multi-site syntheses^{60,61}.

We also added a paragraph following the above paragraph that discusses potential effects on stand-scale mortality and processes such as self-thinning (lns 459-467, tc542-550).

Also in a species comparison, fast growing taxa such as poplar, aspen, but also birch and maple are shorter lived than slower growing taxa such as oaks or beech, the combination of fast growth and long life does not seem to exist in tree species, perhaps due to the tradeoff described by Loehle, or because there is a maximum species-specific tree height, or at community level, because of the distinction between r and K strategists. When r strategist tree taxa would profit from CO₂ enrichment und suppress K strategists, the carbon stock would decline.

Loehle C (1998) Height growth rate tradeoffs determine northern and southern range limits for trees. *J Biogeogr* 25:735-742

The following paper is a rather trivial statistical exercise, but may contain references for the vigor/age/height tradeoff (I did not check):

Stahl U, Reu B, Wirth C (2014) Predicting species' range limits from functional traits for the tree flora of North America. *Proc Natl Acad Sci U S A* 111:13739-13744

The following review covers major US forest tree species:

Purves DW (1989) The demography of range boundaries versus range cores in eastern US tree species. *Proc R Soc B* 276:1477-1484

The r versus K life history strategies (MacArthur and Wilson, 1967) or the growth survival trade-off (Wright et al., 2010) are clearly well established ecological theory and we agree that should CO₂ alter the competitive balance among the species exhibiting these strategies then there would be long term consequences for the terrestrial carbon sink. However, the manipulation experiments in our analysis cannot be used to address the questions concerning these longer-term ecological processes. Please see our more extensive response to the reviewer's similar point below.

Reviewer #2 (Remarks to the Author):

The paper provides an interesting and frankly very critical, thorough and smart analysis of decadal forest biomass response to atmospheric CO₂. The Authors analyze four sites across a large climate gradients that have been exposed to ~10 years of enriched CO₂. The analysis integrates these four sites with appropriate statistical modeling, while tying in simple concepts of biomass accumulation, but also confronts forest models and/or land surface model. The major findings are that forest biomass gains more under higher level of CO₂. Analysis of biomass gains show relationship with net primary productivity and wood allocation. Models are not capable of capturing the breadth of responses across sites, due to the lack of being able to correctly reproduce one or several factors that contribute to the CO₂ response (NPP, NPP response or the response of vegetation increase in response to NPP).

The reviewer's support is very much appreciated.

While a very nice and thorough analysis, a few questions remain.

Novelty of the CO₂ response finding: The FACE experiments have been running for a long time and quite a large body of literature exists detailing biospheric response to atmospheric CO₂, including NPP response, growth responses etc. It is not clear, how this study relates to those earlier findings. For example, to what degree is the number of 22.6 % different than the NPP increase found in Norby et al., 2006? Yes, you translated the NPP response into a biomass increase, and yes you showed that wood allocation increases with NPP? What does that mean to the interpretation of earlier results?

We have added some additional text to the discussion in the manuscript. Some text deals specifically with the 23 % finding from Norby et al. (2005) (lns 331-339, tc360-369):

The CO₂ stimulation of annual NPP observed in this study is consistent with the 23 ± 2 % median annual increase calculated from a linear regression of elevated versus ambient annual NPP (at Rhineland, POPFACE, ORNL, Duke) but a shorter time-period (1-6 years depending on the site) (Norby et al. 2005). Mixed-model analysis, allows estimation of the response of the statistical population and is the reason the uncertainty in this study is larger, 6 % versus 2 % in (Norby et al. 2005). (Norby et al. 2005) showed that across sites and species, the annual NPP response to CO₂ was greater at higher ambient NPP. We did not observe that effect at the decadal scale.

A key finding of our study is that decadal biomass increment was increased by 29.1 ± 11.7 %.
...

We have also extensively edited the discussion and removing the political context, the soil carbon discussion, and other places suggested by the reviewer. We hope this had made the interpretation of the results clearer.

It is not super clear to me why the biomass increase is linear to cumulative NPP when wood allocation is a first order response to it? Wouldn't your finding of a positive fw_b lead to a (albeit small) quadratic increase?

The reviewer is correct, the relationship of biomass increment to NPP must be quadratic in the way we have posed it. See equations M1-M9 that have now been brought from the supplement to the main methods. However, as we have analysed it, the statistical relationship to NPP is linear. Given the data, fitting a quadratic term to the statistical model would not perform better than a linear model. If we forced the intercept through zero, then a quadratic would likely be a better fit, though this would presuppose a biomass response if NPP responds to CO₂ and this connection is what we wanted to test.

We recognise that our presentation here was not completely clear, and as the reviewer points out this has caused some confusion. We now refer to the quadratic in the abstract (see response to specific comment below). And we modify text in the results to (lns 357-358, tc394-395):

While approximately linear, the combined direct effect of increased NPP by CO₂ and the indirect effect of increasing NPP on fW suggests that the actual relationship of Cinc_{veg} to cNPP is quadratic.

We hope that this has made the relationship between the statistical models and the more process based, underlying hypothesis clearer. If not, we would be happy to work on this further.

It is not clear to me why the authors choose to include a lengthy discussion of soil carbon response and aggrading vs. non-aggrading systems. While this offers some context, this discussion brings in often information that is less relevant to the paper. I suggest considerable shortening starting at line 372. Similarly, the last sentence of the abstract also seems to distract from the major finding and can be shortened. These additions seem as the authors want to make sure that there is more to the story of CO₂ and climate response.

The terrestrial carbon sink context is the reason for these discussions, and is probably not as necessary as we first thought (indeed some co-authors advocated removing this context in the original draft). We have cut substantial passages of text from the manuscript, including the soil carbon paragraph, and the more political, terrestrial carbon sink statements made on the final line of the abstract and the final paragraph of the main text.

The aggrading vs. non-aggrading discussion was intended to preempt some of the criticisms leveled by reviewer 1. Given the strong statements from reviewer 1 and our general agreement that the successional context is important, we maintain and expand this context in the introduction, results and discussion.

I like very much the inclusion of the models, and particularly the decomposition of the response into several factors, these are critical pointers to move models (that should be capable to capture a CO₂ response) forward.

The reviewer's support is again appreciated. Our aim is to synthesise data in a way that is useful to understanding process and to evaluate models in the context of process and improved representation of those processes.

Detailed Comment:

Abstract: Population-level results. This becomes clear when reading the method section of the paper. But from the abstract it almost reads like the response is evaluate based on the population of trees. I suggest revising this.

The abstract has been edited to remove this potentially confusing terminology. Now reads (lns 58-63, tc62-68):

Here, mixed-model analysis of data from four early secondary-successional ecosystems dominated by woody plants shows that a decade of experimental CO₂ enrichment increased NPP by 22.9 ± 6.1 % and decadal biomass carbon increment by 29.1 ± 11.7 %. Total biomass increment increased in quadratic with cumulative NPP (approximately linear over the range of observed NPP), via a direct effect of NPP on biomass increment and an indirect effect of a larger fraction of NPP allocated to wood as NPP increased.

L63: Total Biomass increment increases linearly with cNPP, does this not

contradict your finding of a positive f_w , and the fact that the linear relationship has a negative intercept? Please explain in the main text what exactly is meant.

Yes, positive f_w contradicts this statement. The relationship is quadratic, though the quadratic can be approximated with a linear relationship over the range of NPP values in the study. We have edited the line in the abstract as follows (lns 60-63, tc64-68, and see text quoted in previous response).

L68: second half of the sentence starting with albeit: This is not a finding of this study, I suggest to delete. I understand that there is a need for caveats, but this is probably better placed in the conclusion.

Agreed, this text to the end of the abstract has now been deleted.

Line 75: Suggest to write: "This uncertainty arises PARTLY due...", there are other uncertainties

Agreed, there are other uncertainties. We added "in large part" as much of the uncertainty can be ascribed to how these processes are represented.

L 86: typo "experiment THAT lasted"

Thanks, corrected.

L133: typo: explanatory

Thanks, corrected.

L285: First sentence: This has not been tested, but assumed in the model (I do not oppose that assumption). However, it reads like as this has been statistically tested.

Agreed. Sentence now reads (lns 361-363, tc399-401):

A potential explanation for the linear relationship between f_w and cNPP (Table 2) is that allocation to resource acquisition organs (i.e. leaves and fine roots) is a higher priority at lower rates of NPP.

L322: Not clear whether "predicted" belongs to the statistical model devised here, or the mechanistic models.

In this sentence predicted refers to the mechanistic models. The text now reads (lns 389-390, tc447-448):

At all sites, large variability in f_w predicted by the model ensemble propagated through to large variability in the predicted biomass retention rate.

L327: Wood allocation is a linear function of cNPP. This is not clear. Is it the fraction allocated to wood that is a linear function of NPP (Equation 2), or the amount of NPP that is allocated to wood is constant (absolute vs. relative terms)? This is a bit confusing and I hope this can be a bit better explained (also together with the statement elsewhere that Biomass increment increases linearly with cNPP).

The former, and we have edited the text for clarity (please see responses to other comments for specific details and line numbers of how we have dealt with the linear vs quadratic inference).

L372: I appreciate the authors acknowledging that this is not whole system C accumulation, but I think this caveat can be considerably shortened since this was not really part of the system.

Agreed, this paragraph has been shortened to two sentences that read (lns 454-457, tc537-540):

If higher growth rates were to increase mortality rates, soil C inputs would be increased. How soil C responds to CO₂ enrichment is an active area of research that must also be considered in analyses of feedbacks between the atmosphere and terrestrial ecosystems⁵.

L383: Also this paragraph has the potential to be shortened (and perhaps put in a single paragraph combining all the caveats and limitation of the study.

We have edited this paragraph to be the concluding paragraph that summarises the limitations, the key findings, and priorities for future research. The paragraph now reads (lns 469-478, tc552-570):

CO₂ responses conditional on successional stage indicates that the direct response of the atmospheric carbon sink to increasing CO₂ will depend on the partitioning of forests among successional stages. According to the 2015 Forest Resources Assessment⁶³, temperate forests cover 684 Mha of Earth's surface. Of this temperate forest cover, 60 % is naturally regenerating, secondary forest and 22 % is plantation forest. Even primary forests have gap-dynamics, caused by natural disturbance and tree mortality, that can be thought of in terms of secondary succession. Predictive understanding of terrestrial ecosystem biomass responses to increasing CO₂ will require synthesising understanding of CO₂ interactions with other production-limiting resources; the relationship between wood allocation and NPP; and the interactions between growth, mortality, and self-thinning through the stages of forest succession into a coherent, mechanistic theory. This should be a terrestrial C cycle research priority.

L398: Again, another paragraph for shortening potential.

We have deleted this paragraph.

Reviewer #3 (Remarks to the Author):

The purpose of this study is to determine if net primary productivity is limited by carbon and if the biomass increment is significant at decadal time scales in aggrading temperate forests (line 96-97). Methods included both a statistical approach, and an approach using ecosystem models. Ecosystem models were also examined to identify any common areas where the models could be improved (line 213). [As Kaume et al 2014 *New Phytologist* p885 says 'we did not statistically evaluate models against observations because models can easily yield quantitatively good responses for incorrect reasons; thus such an approach typically does not correctly diagnose model deficiencies'] Both of these approaches relied on four experimental sites that involved CO₂ enrichment for a period over ten years.

The authors have used a complex statistical approach to tease information out of the experimental data, and much of the text focuses on the more complex items. It is the lack of information about the basics of the study, or the way the information is presented that makes it difficult to follow the manuscript.

Clarification about the basics of the study, or including the basics, is needed. In the title and several places in the text, the study is about aggrading temperate forests, and yet table 1 clearly lists the KSC site as being 'mature'. Lines 348-356 discuss "the only CO₂ enrichment experiment replicated at the forest stand scale in a mature forest", and the need to understand responses in non-aggrading forests (with the implication that non-aggrading forests are mature - which is what the word mature usually implies: no longer aggrading). Why is KSC included in a study on aggrading forests if the site is mature and therefore possibly not aggrading? Also, in the manuscript KSC is referred to as sub-tropical. How is sub-tropical related to temperate? Reasoning through this, I wonder why KSC is included in this study on aggrading temperate forests because it appears at first look to not be aggrading, nor is it temperate?

We recognise that some of this terminology has potential for confusion and we have edited the manuscript to try to be clearer about our definitions of terms.

Our use of "mature" to describe KSC was intended to indicate that the trees were sexually mature and could reproduce and that the individuals were relatively old (we do not have a clear age estimate for these individuals). The ecosystem at KSC was also "aggrading" (by our original definition of accumulating biomass) as biomass was accumulating in the system due to almost complete above-ground biomass removal by fire in the year preceding initiation of the experiment. In the revised manuscript we revise our "aggrading" classification of these ecosystems as "early secondary successional," and although the plants at KSC are well established, they are rapidly accumulating biomass following frequent disturbance. Adaptation to frequent disturbance does not fall cleanly into the phases of secondary succession. We replace the use of "mature" with "late(r) successional" in the manuscript it is a more relevant descriptor for the study. In Table 1 we replace the "age at start" column with a "time since disturbance" column.

Classification of KSC as both temperate and sub-tropical is not inconsistent. KSC is within the temperate latitudinal zone, and the sub-tropical climatic zone of Köppen-Geiger is a sub-classification of temperate (Kottek et al., 2006). In the revised manuscript, we continue to classify all of these sites as temperate as they are within the temperate zone. We are more explicit about using the Köppen-Geiger classification when describing the sites, placing Rhinelander in the hot/warm-summer without dry season cold (humid continental) climate; and Duke, ORNL, and KSC in the hot-summer without dry season temperate (humid subtropical) climate (Kottek et al., 2006).

The term 'semi-natural' is occasionally used in the text which gives the reader pause. What is meant by this? Of the four sites, three are plantations, and the one that is not a plantation is: KSC. Here is a third item where this site is different (not to mention it is the only site which is an OTC rather than a FACE). Why does this site belong in this study? A paragraph stating why these sites are included in this study, that these are the only 4 sites in the world

that meet the criteria for inclusion in this analysis, is necessary.

Thanks for pointing this out. “Semi-natural” is not particularly helpful, what was meant by it was unmanaged during the experiment. We have removed the term “semi-natural” and have been clearer about our choice of sites. The text now reads (lns 84-96, tc93-107):

Here we synthesise NPP and biomass responses to CO₂ from the four ecosystem CO₂ enrichment experiments in ecosystems dominated by woody plants that lasted a decade or longer, were unmanaged during the experiment, were replicated at the ecosystem scale, and quantified all components of NPP and biomass (Table 1). ... Our selection criteria restrict the analysis to temperate ecosystems dominated by woody plants (woody ecosystems), in the early phases of secondary succession.

and (lns 511-514, tc615-618):

Our selection criteria excluded three woody ecosystem CO₂ enrichment experiments: POPFACE, WebFACE, and EucFACE from the analysis, as they were either managed, did not quantify NPP and biomass, or had been running for just five years, respectively. We have a maximum of 11 years of data for each experiment.

Based on the initial results, however, the experimental site that seems to not fit well is ORNL. It is also the only site that does not seem to have a 0.55 biomass production rate, which appears to be more of a biomass production ratio. (for every unit of NPP, what is the resulting biomass increment?) I wonder what the results for ORNL would look like if the 0.55 was imposed on the calculations?

That is correct, in terms of the biomass increment ($C_{inc_{veg}}$) to cumulative NPP relationship, ORNL is the site most different from the other three. The results from the model selection exercise suggest that the best model includes a random effect of site not just on the intercept but also on the slope. Therefore, all sites had different biomass production rates (Table 2 Model 3). The reviewer’s final suggestion is interesting, though it is not completely clear which calculations the reviewer is referring to. It is also unclear how this would be done in a mixed-model analysis. To some degree, the distribution of the bootstrapped model agrees with the reviewer that the biomass production rate could be higher at ORNL – the median of the bootstrapped distribution has a higher biomass production rate than the model fit to all the data (Figure S1). We have no basis to conclude that with greater replication, ORNL may have had a stronger biomass response.

We agree that the “biomass production rate” is similar to the ratio but it is not exactly the ratio. The biomass production rate, as we define it, is the differential of the $C_{inc_{veg}}$ to cNPP line. In the case where the line goes through the intercept the rate and the ratio are the same, but where the line does not go through the intercept (as the analysis suggests), the rate and the ratio are different.

Biomass increment (line 125-126) was measured as the ‘difference between biomass in the final year minus biomass in the initial year of the experiments’. Please clearly define succinctly what components are included in biomass increment. Going through the additional methods, it appears this includes coarse roots, and all live vegetation (includes understory but does it include foliage of all live vegetation?). It appears that a few measurements were taken, and then biomass was calculated using formulas. There may be measurements, but biomass is not measured, it is estimated or modeled. Also, by definition, any vegetation that died over the time period simply drops out, meaning if there was a lot of mortality biomass increment could be negative.

Thanks for this comment. We were more explicit about the methods in the supplement but not in the main paper. We have now brought all of the methods that were in the original supplement into the

methods of the main manuscript. We also augment these methods to address the points made by the reviewer. We agree that biomass increment could be negative. The additional methods text reads can be found on lns 516-544, tc620-648.

In Figure 1, in the inset in the top left corner, could mortality be the reason for the small increase in biomass increment as compared to cumulative NPP? (One would think so, by definition.) To truly understand what is being attributed to as a biomass increment response to CO₂, the manuscript needs to clearly and concisely articulate the simple information/calculations needed to come up with those estimates. What biomass equations were used to estimate biomass increment? Were these equations used to estimate NPP? Was NPP measured at each site each year, or are NPP measured/estimated across some years? In the additional methods section, I can pick up some of this information but having important factors summarized in a succinct table for easy comparison would be very useful to help the reader understand possible fundamental differences in results from each site.

Across all sites, the primary reason for the smaller increase in biomass compared with cumulative NPP is the allocation of that additional NPP to high (roughly annual) turnover tissues. Tree mortality is a component of the shortfall, but one that makes less of a contribution.

A table summarising all of these methods is a good suggestion, thanks. We add a table to the methods that explicitly details how these measurements were made at each site. To respond to several of the reviewer's questions here: NPP was measured consistently in every year at three sites. The only site that did not measure NPP consistently in all years was Rhinelander, where leaf and fine-root production were not measured in 1998-2001. We assume the reviewer is referring to allometric type equations with "biomass equations." Wood and coarse root biomass were calculated from allometric equations applied to DBH and other variable measured on an annual basis. The wood and coarse root components of NPP were calculated using the annual increment in their standing stocks on a per tree basis (at Rhinelander, ORNL, and Duke). This means that mortality is not included in woody NPP and that woody NPP has a minimum of zero.

In compiling this table, we decided that to be consistent with the other sites the herbaceous component of NPP and biomass at Rhinelander should be omitted. This minor revision to the dataset made a small difference to the calculated values from the statistical models and is the reason the quantified values have changed slightly throughout the manuscript.

If this was a study focused only on ecosystem models and processes the above information may not be as important. But these items matter in an empirical approach.

We completely agree, thanks for bringing these points up and the study is more transparent as a result. We are committed to open and transparent science.

This study could be vastly improved with one close read by an author, ensuring the basics of the sites and study were well-covered and consistent throughout. Something as simple as using a spell-checker would be a nice touch for the reader, given there are random mis-spellings throughout which distract (For example, line 133 expanatory; line 158 reationship; line 369 increased). Line 320-321: am wondering if the term 'highly uncertain' is more accurately 'highly variable'?

Apologies for this inconvenience. The lead author takes responsibility for the poor spelling. These mistakes have now been corrected and spelling in the revised manuscript checked thoroughly.

The purpose of the study needs to be improved, by stating it explicit and crisp. Be clear why these four sites are all appropriate to include - were any sites omitted that could have qualified? (such as in Europe)? There is a discussion

about biases that is confusing. Does the statistical technique induce bias? Is bias being used more in terms of the ecological models? Is it 'true' bias in the statistical sense, or is the term 'bias' simply being used to indicate poor model fit, perhaps due to throwing data together which a statistician would omit as outlier? Or an ecologist may omit because the site does not meet the description of sites under study?

We have attempted to improve the crispness of the objectives of the study, and the details of the site selection (please see response to the comment above about site selection). With regards to objectives, we hope the first sentence of the abstract sets the scene crisply (lns 57-58, tc61-62):

Stimulation of photosynthesis by increasing atmospheric CO₂ can often result in increased net primary production (NPP), but at longer timescales, may not result in increased biomass.

To be more explicit about the studies objectives, we have added a final paragraph to the introduction that details these objectives (lns 113-125, tc127-139):

We focus on four inter-related questions: in woody ecosystems that are in early phases of secondary succession, 1) is NPP carbon limited at the decadal time scale? 2) If so, is biomass increment carbon limited at the decadal time scale? 3) Is there a relationship between biomass increment and cumulative NPP? 4) If so how does experimental CO₂ enrichment affect the relationship? Evidence for carbon limitation of NPP and biomass increment is defined simply as an increase in the respective variable at elevated CO₂ concentrations. By reducing the influence of transient effects and inter-annual variability, the decadal timescale is critical to the inference of any potential NPP responses to elevated CO₂ and if NPP responses are retained as biomass at longer, more global change relevant, timescales. The secondary successional stage of these ecosystems is also an important factor in interpretation of any CO₂ responses. The shifts in resource availability associated with secondary succession¹⁶⁻¹⁸ are likely key factors influencing the CO₂ response and evidence suggests that CO₂ responses reduce under tightening resource coupling associated with succession¹⁹ and with the age of individuals²⁰. Thus we also interpret our results in the successional context.

In summary, the complex approaches used here may be solid, but improved explanations are needed, especially on the basics. The topics being discussed are quite important, but the manuscript needs to be improved.

We thank the reviewer for their support and hope that our edits and additions in response to their thoughtful comments have improved the clarity and completeness of the manuscript.

References not cited in manuscript:

- Bugmann, H., Bigler, C., 2011. Will the CO₂ fertilization effect in forests be offset by reduced tree longevity? *Oecologia* 165, 533–544. <https://doi.org/10.1007/s00442-010-1837-4>
- MacArthur, R.H., Wilson, E.O., 1967. *The Theory of Island Biogeography*. Princeton University Press, Princeton, NJ, USA.
- Kottek, M., Grieser, J., Beck, C., Rudolf, B., Rubel, F., 2006. World Map of the Köppen-Geiger climate classification updated. *Meteorologische Zeitschrift* 15, 259–263. <https://doi.org/10.1127/0941-2948/2006/0130>
- Wright, S.J., Kitajima, K., Kraft, N.J.B., Reich, P.B., Wright, I.J., Bunker, D.E., Condit, R., Dalling, J.W., Davies, S.J., Díaz, S., Engelbrecht, B.M.J., Harms, K.E., Hubbell, S.P.,

Marks, C.O., Ruiz-Jaen, M.C., Salvador, C.M., Zanne, A.E., 2010. Functional traits and the growth–mortality trade-off in tropical trees. *Ecology* 91, 3664–3674.
<https://doi.org/10.1890/09-2335.1>

Reviewers' comments:

Reviewer #1 (Remarks to the Author):

Comment on Walker et al. NCOMMS-17-24567

I had commented this paper earlier and find this revision is meeting the requests that came up in the first round. The authors adopted successional stage as key criterion to explain part of the results. Also the second focus, that on fW (fraction of C ending up in wood) comes across well in the current version (though partly repetitive). The wording had improved a lot and the questionable political issues had been removed. I am impressed by how much effort the authors invested to improve the analysis and its presentation. With the below minor suggestions, I think this can be published

46 The key word 'terrestrial carbon sink' does not seem appropriate, because the study does not and cannot say anything about sink effects (an ecosystem property including soils). The paper discusses whether and how a growth stimulation can contribute to this, and that discussion is based on other works. Why not use 'carbon cycle'?

Abstract: Please rethink language. The abstract has 5 sentences. Of these, sentence 2 and 5 are over-long, complex, and not really clear. For a reader who has not read the main text, it remains cryptic how something that is independent of CO₂ can be predicted from a CO₂ response of NPP. I know what you want to say, but this statement is not well phrased. I found line 204-206 well phrased, if abbreviations can be avoided....

135 I am afraid many readers will take 'steady-state' as 'constant'. Perhaps, an explanation in () can help? (e.g. annual leaf and fine root turnover operating around a stable mean)

146 I do not think ORNL is special in that respect. I guess all four sites would have responded to N-fertilizer addition by faster growth? I remember earlier works from Duke FACE that made a strong case for N-limitation.

165 I would not phrase this as an 'adaptation'. All species that are able to resprout after cutting would produce suckers and extraordinary initial above ground growth (a rich coppicing literature)

181 I do not wish to stir up things at this point, but responses during these early years are most likely setting the scene for the following years. I wonder what happened if one stopped CO₂ enrichment after year three? At that point in time each system had established a CO₂ driven initial response (a resulting biomass 'capital') reflecting the early nutrient condition. Adding a constant 'interest' to different 'capital' will cause the two systems to depart from each other over time, no matter whether CO₂ is on or off. So these early conditions bear the greatest potential to explain the outcome. If I understand correctly, ORNL was the only system that had arrived at steady state LAI before CO₂ treatment started (close to 6, a typical max LAI for temperate deciduous forest), and all other stands started with a low LAI, in fact, KSC and RHIN with close to zero. Duke is difficult to explain because the canopy is quite open (a strong 3D roughness, that makes LAI difficult to measure optically, because the homogeneity-rule is objected, and needle retention and thus, litter fall may vary from year to year). The large Figure with all that info was very helpful.

193 The net biomass increment...

193-194 That sentence simply says that all 4 stands grew.... Who would have expected otherwise?

195 I do understand why you coined Cinc-veg. Yet, the difference to the popular NEP is not made obvious to readers. The main reason is that you exclude soil C. All your calculations are built upon

growth and biometry data.... You could employ NEP with a caveat in methods... Would make reading so much easier... Maybe NEPv could do? Just an idea...

204 I like that clear statement

217 time? ...perhaps duration, time laps, period?

218 On longer time scales, vegetation turnover is determined....

232 ...a biomass response to CO2 enrichment the neg...

238-239 Couldn't this be species specific? Some taxa are more 'woody' others more 'weedy'. Also wood density and thus, tissue duration differs (think of scrub oak and aspen)

258 ...a biomass response to CO2, Rhine... I think the fw story adds a good deal of basic science to the analysis that goes beyond the CO2 fertilization issue.

320 please replace 'forest'. KSC is scrub as the name says. This scrub system simply does not deserve the word forest, nor do the young stages of the aspen experiment.

328 ...evidence that these early...

329 ...increment were carbon....

I suggest to be careful with the generalization given the very special settings of these different experiments.

339 my comment to line 181 applies here. Had these 29% already been established very early in the experiment and just be retained? Can you bring a bit light into that 'compound interest' story?

347 the central role of the nutrient cycle for the C cycle was recently shown for grassland by Reich et al in their April Science article.

357-360 Isn't that a repetition?

363 NPP is expressed per unit land area. Hence regrowth or seedling/cutting growth from near to nothing such as at KSC and RHINE would yield very little NPP at very high individual growth rates till canopy closure. I rather think this is related to stand age. Once leaves and fine roots 'filled' the space, there is only wood increment left (both below and above ground) to incorporate more C in biomass per unit land area. It needs no 'strategy' in the sense of a 'priority' to explain that. This is the very nature of steady state (since coined by Odum).

392-407 Since the nutrient availability most likely controlled C incorporation at all 4 sites, it is unfortunate that such nutrient data are not provided. Whatever modelers assumed in that respect will have driven model output.

402 ...models with an N cycle.... based on which input data? So, some of the models employed had no nutrient cycle? How can growth be modeled without nutrients?

406 refer to Reich et al Science 2018, they showed exactly this.

426 However.... suggest to remove that sentence. The experiments (and models) presented here have the same limitation. These central European forests are likely to operate at N saturation given the enormous N-deposition rates as was suggested by Schleppei et al. (2012) in Global Change Biology.

Why not refer to Sune Linders OTC experiment with tall spruce? They showed that the CO2 response could be switched on or off by nutrient addition.

Sigurdsson BD, Medhurst JL, Wallin G, Eggertsson O, Linder S (2013) Growth of mature boreal Norway spruce was not affected by elevated CO2 and/or air temperature unless nutrient availability was improved. *Tree Physiology* 33:1192-1205

438 ...NPP with N at our test sites.

452-454 I find the handling of this matter imbalanced. Ref 54 and 55 provide info on undisturbed systems, 57 is a model with an emphasis on simply showing the importance of mortality and NOT a vigor-lifespan relationship. 58 and 59 talk about fire driven systems, where fast early growth creates a protective bark that better resists fire. These papers do not refer to biological life expectancy. Sure, all this matters, but this § reflects bias ('a broader body'). I do not see that 'broader body' for the current question.

455 mortality to increase soil C inputs? Not necessarily. The effect could be neutral or even feeding back negatively on growth because of nutrient trapping in debris.

All these are minor issues of wording and literature treatment. My comment to line 181 and 339 may deserve a few remarks/numbers, given the systems differed strongly in that respect. The response during this initial stage may, in fact, hold much of the answer to the final outcome.

Reviewer #2 (Remarks to the Author):

The authors have worked diligently to address the reviewers' comments. Thank you! A few questions remain (or came up with the reading of the revised manuscript).

Interest to others in community and wider field: The findings are definitely in the interest of the community and in the general field of climate change and global biogeochemistry. It summarizes nicely and in a fairly comprehensive way the main results of FACE experiments after a decade or so of experimentation. I also think the way the used statistical model is state of the art and the inclusion of models tasks to quantify those feedbacks for future climate feedback enhances the discussion. While I am definitely interested to see this work published, I cannot make up my mind whether this is sufficiently novel for a much broader community of natural sciences.

Main result reporting: I am wondering whether the percentage NPP response is actual the right metrics to report in the abstract and as a main conclusion. The work suggests that the baseline NPP is shifting, while the response to higher CO2 (in absolute terms) is much more indicative. Table 2 suggest that across all sites the NPP response is 0.164 ± 0.031 (kg m⁻²). If then expressed as a percentage, it is then divided by the random effect of each site, diluting the precision.

New discussion on forest development: I appreciate the inclusion and the discussion of the stages of regrowth in these sites. I am wondering, though, whether they could be even more integrated, by for example discussing the allocation to wood across the different sites. Is there an expectation that a shift from one phase to the next also leads to a shift in wood allocation? Or alternatively, is there a potential change in retention rate along the successional development stages?

Cinc/NPP relationship: I understand that the fraction allocated to wood is a modeled variable as it relates Cinc with NPP. However, I think that in each of the sites, wood increment has been measured in order to obtain NPP. Can the estimated allocation to wood via Cinc-NPP relationship

be verified by actually calculating the 'real' (i.e. estimated from wood increment) data? There may be a need to make certain assumption (no mortality, for example), but I don't think these assumptions would be more tenuous than the empirical fw (Cinc/NPP) function in the existing manuscript?

Model data discrepancy and its discussion: It becomes clear that one of the failures of models to capture the CO₂ effect is their failure to predict wood allocation at these sites. Many models currently have very simple rules for wood allocation (fixed, pipe model, and others). However, it is less clear how models can improve. As I understand fWa has site specific random effects. Yet models need to be able to predict these in order to capture the CO₂ effect. I would appreciate potential discussion points (does the literature suggest some relationship? – Is this something that needs further evaluation and what kind of data could help with that?).

Minor and editorial stuff:

L147 I believe it should be N_{15} instead of N_{15}

L157 What does "this value" refer to? Resource acquisition phase, saturation? Please clarify

L158 "However, this evidence for expanding resource acquisition volumes was not apparent in the NPP data". How would expanding resource acquisition present itself in the NPP data?

L163: "the trend was highly variable" – meaning no trend?

L236: Isn't 3) a logical consequence of 1) and 2)? That is the third prediction is not independent? A simple therefore before "3)" might just be enough to clarify.

Reviewer #3 (Remarks to the Author):

The authors' responses to previous comments are generally on target, and appreciated. With the improvements, it is now easier to see the issues that need attention.

The importance of the comparability of the data cannot be emphasized enough when only four sites are available. It is not necessary to include in the main text the entire Table 4 from the supplementary information, but it is important to state in a paragraph or so whether the way these data were collected/measured may influence the results. Table 4 is somewhat difficult to read, but it looks like coarse roots are generally calculated at all sites as a function of diameter-based allometric relationships. KSC looks to have had more attention to the root-related estimates, with both soil cores and mini-rhizotrons, and an isotopic tracer approach was used for a turnover rate. How do authors know that differences between the different pools on these sites, especially roots, is not largely related to the methods for producing the data/estimates? A summary paragraph or two in the text to convince the reader the data from these sites is comparable in spite of different collection techniques and in spite of perhaps using more localized biomass equations would be useful. And move Table 4 back to supplemental, and reference that it was there.

Estimating NPP from above-ground inventory data is an accepted technique (see Jenkins et al 2001 for example). A study like that one makes clear how much NPP is related to biomass increment by definition. Here, the wording of the research questions need to be reconsidered, especially the question "is there a relationship between biomass increment and cumulative NPP"? This an obvious answer: yes, by definition. Mortality will affect standing live biomass increment. If one is examining a graph of only biomass increment and only cumulative NPP plotted against each other those variables may seem unrelated, but it is just that mortality is being ignored. Consider modifying the questions for clarification. The abstract indicates one question is: is the relationship of wood allocation to NPP affected by atmospheric CO₂ level? Is this one of the questions? It seems as if I am still reading an early version of a manuscript, and that the ideas in the text need to be made more clear and crisp.

Another possible issue and why a focus on data is important: if CO₂ affects tree growth such that the taper of the tree bole is affected, such as, trees have a larger diameter but do not proportionately increase in height or proportionately do not increase in diameter at different heights up the tree, then using the same biomass models for different time periods would ignore that effect. The independent relationship of wood allocation to NPP may be masked without having more precise estimates of wood production. (an example related to crown dynamics is Valentine et al. (2013)).

Turning to the ecosystem models part of the study: It is not clear why it is included here. Lines 278-279 indicate that they were used "to simulate these four CO₂ enrichment experiments and identify any common areas where models might be improved." There is no listing of common areas where models might be improved. Examining ensemble results usually is not that helpful when considering model improvement because the results are all muddled. Instead (back to lines 320-321) the modeling seems to be included to investigate ecosystem scale effects. For example, as presented in lines 288-290 ("Partitioning the modelled Cincveg response to CO₂ enrichment into the cNPP response to CO₂ and the CO₂-independent biomass retention rate, as described for the observations, allows us to identify which of these processes was responsible for the model bias."), it looks like the models are being examined to test how they assume partitioning occurs, and comparing results to these data, which includes some modeled pools or pools estimated using different methods so back to the importance of the assumption of data comparability. Perhaps the main confusion is the text (lines 278-279) that indicates "any common areas" for improvement are doing to be identified. It sounds as if the reader should expect a list. But the models appear to be introduced to investigate a specific phenomenon.

Over the years, the FACE program studies have focused on crucial questions, and as a result are of interest and importance in the scientific literature. Strictly from a research and a statistical point of view, in general, making inferences based on four ecosystem sites to address questions on effects of CO₂ to apply to all temperate woody ecosystems seems like a stretch. Lines 320-321 says "The scale of these FACE and OTC studies (ecosystem and decadal scales) allows analysis of forest responses to CO₂ enrichment at scales at which the carbon cycle becomes relevant to climate change." It is a stretch to have one data point (meaning the four sites each cover one decade) and say the results apply to decadal scales. Nonetheless, this publication would be typical for FACE studies given there are no alternative sources of data, and the results may be of interest and be very meaningful to that scientific community.

The text needs tightening up though because there appear to be inconsistencies throughout as to what the questions are being investigated (see abstract—are all the questions answered directly there?), and why certain methods are used. The manuscript needs more thoughtful attention to be understood by the broader scientific community.

Jenkins, Birdsey, et al. 2001. Biomass and NPP Estimation for the Mid-Atlantic Region (USA) Using Plot-Level Forest Inventory Data. *Ecological Applications*, 11(4), pp. 1174-1193

Valentine, HT. 2013. Crown-rise and crown-length dynamics: application to loblolly pine. *Forestry*, 86, (3), 371–375

Response to Referees: manuscript NCOMMS-17-24567B

Title: Decadal biomass increment in early secondary successional woody ecosystems is increased by CO₂ enrichment

We appreciate the supportive and constructive nature of the reviews, and we thank the reviewers for their time and efforts to improve our analysis. We have carefully considered each of the reviewers' comments and incorporated the majority of them in the revised manuscript which, in our view, has strongly improved the study. In responding to their comments we feel the study is better framed and more readily interpretable. For ease of reading our responses are in blue font, with direct quotations from the manuscript in bold blue font.

Reviewers' comments and detailed responses:

Reviewer #1 (Remarks to the Author):

I had commented this paper earlier and find this revision is meeting the requests that came up in the first round. The authors adopted successional stage as key criterion to explain part of the results. Also the second focus, that on fW (fraction of C ending up in wood) comes across well in the current version (though partly repetitive). The wording had improved a lot and the questionable political issues had been removed. I am impressed by how much effort the authors invested to improve the analysis and its presentation. With the below minor suggestions, I think this can be published

We thank the reviewer for their positive summary. We did indeed try hard to consider and meet the reviewer's original comments and requests. We agree that successional status is a useful way to frame the question of CO₂ fertilisation.

46 The key word 'terrestrial carbon sink' does not seem appropriate, because the study does not and cannot say anything about sink effects (an ecosystem property including soils). The paper discusses whether and how a growth stimulation can contribute to this, and that discussion is based on other works. Why not use 'carbon cycle'?

We have made the change.

Abstract: Please rethink language. The abstract has 5 sentences. Of these, sentence 2 and 5 are over-long, complex, and not really clear. For a reader who has not read the main text, it remains cryptic how something that is independent of CO₂ can be predicted from a CO₂ response of NPP. I know what you want to say, but this statement is not well phrased. I found line 204-206 well phrased, if abbreviations can be avoided....

We agree. Thanks for the opportunity to address these issues and the suggestions. The revised abstract now reads:

Stimulation of photosynthesis by increasing atmospheric CO₂ can increase net primary production (NPP), but at longer timescales, may not necessarily increase plant biomass. Here, we analyse the four decade-long CO₂ enrichment experiments in woody ecosystems that measured total NPP and biomass. Mixed-model analysis shows that CO₂ enrichment increased biomass increment by 1.05 ± 0.26 kg C m⁻² over a full decade. This biomass response was predictable with a combination of the NPP response to CO₂ (0.16 ± 0.03 kg C m⁻² y⁻¹) and the CO₂-independent biomass retention rate (slope of the relationship between biomass increment and cumulative NPP ; 0.55 ± 0.17). An ensemble of terrestrial ecosystem models failed to predict both terms correctly, but with different reasons at each site. CO₂-independence of the biomass retention rate highlights the value of understanding ambient conditions, which were linked to successional stage, for interpreting CO₂ responses.

135 I am afraid many readers will take 'steady-state' as 'constant'. Perhaps, an explanation in () can help? (e.g. annual leaf and fine root turnover operating around a stable mean)

We also agree. We have removed this section of text and limited discussion of the Bormann and Likens scheme of secondary succession to the first three phases as these are the ones relevant to these four sites. The relevant point with regards to 'steady-state' ecosystems, in our opinion, is that steady-state is really a mosaic of the other three phases and is a state that applies to the scale above the patch scale (where patches are defined at the scale of a stand replacing disturbance event). We have substantially edited the text in the results that describes the secondary successional stage (lns 145-193). This has not changed in terms of content but is clearer and easier to follow.

146 I do not think ORNL is special in that respect. I guess all four sites would have responded to N-fertilizer addition by faster growth? I remember earlier works from Duke FACE that made a strong case for N-limitation.

We agree that the other three sites would likely have responded to N fertilisation. Indeed, it was shown that plot-to-plot variation in NPP at Duke is determined by variation in N fertility (McCarthy et al., 2010), and this has been shown at other sites as we discuss on ln 426. Our point is *not* that ORNL is special in the regard that it responds to N fertilisation, but that it is special because in both treatments there was a peak in NPP followed by a decline over the course of the experiment. And it is the decline that was avoided by N fertilisation. This is indicative of progressive N limitation in the ambient treatment, that was exacerbated by CO₂ fertilisation. This suggests that ORNL was on the cusp of the aggrading and transition stages as defined by Bormann and Likens. The key phrase in the text reads (ln 170):

... [ORNL] was experiencing progressive nitrogen limitation under *ambient* CO₂ conditions.

165 I would not phrase this as an 'adaptation'. All species that are able to resprout after cutting would produce suckers and extraordinary initial above ground growth (a rich coppicing literature)

Agreed, many broadleaved species are able to resprout vigorously. Given the frequency of disturbance events at KSC it is likely that the scrub oaks are especially adapted to resprouting, as

indicated by the low fW_a and therefore high below-ground allocation. That said, the reference to adaptation at this point in the text is not really necessary so we have removed it. The sentence now reads (187-193):

This rapid post-fire response could have been supported by nutrients released from fire and large below-ground reserves indicated by 5-10 fold higher fine-root biomass than at the other three sites. The very high NPP in the early years of this experiment suggests that growth was uncoupled from resource availability, with coupling to resources increasing as NPP declines through the experiment. KSC does not fit cleanly into the scheme of Bormann and Likens as the disturbance did not kill the trees and the stand is recovering from live below-ground organs and not from seedling re-establishment.

181 I do not wish to stir up things at this point, but responses during these early years are most likely setting the scene for the following years. I wonder what happened if one stopped CO₂ enrichment after year three? At that point in time each system had established a CO₂ driven initial response (a resulting biomass 'capital') reflecting the early nutrient condition. Adding a constant 'interest' to different 'capital' will cause the two systems to depart from each other over time, no matter whether CO₂ is on or off. So these early conditions bear the greatest potential to explain the outcome. If I understand correctly, ORNL was the only system that had arrived at steady state LAI before CO₂ treatment started (close to 6, a typical max LAI for temperate deciduous forest), and all other stands started with a low LAI, in fact, KSC and RHIN with close to zero. Duke is difficult to explain because the canopy is quite open (a strong 3D roughness, that makes LAI difficult to measure optically, because the homogeneity-rule is objected, and needle retention and thus, litter fall may vary from year to year). The large Figure with all that info was very helpful.

We are glad the reviewer likes the new Figure 1. We also think it is useful for comparing these sites side-by-side. The reviewer is invoking a relative growth rate argument which has been discussed in detail in the literature some time ago (e.g. Norby et al., 1999, 1995). But the reviewer mistakenly considers wood biomass as the capital upon which the “interest rate” is applied. Rather, the capital is leaf area and fine-root biomass (which is why we analysed these ecosystem properties in response to the reviewer’s first round of comments). Indeed this is why the successional context is so important. In an uncoupled state of resource cycling with plenty of space surrounding individuals to expand into, larger individuals have greater capacity for resource acquisition. This allows exponential gains in response to CO₂ (Norby et al., 1992). The reason the stand scale and the successional context is so important is that when above-and-below-ground resource acquisition space is filled a constant interest can no longer be applied to increasingly large biomass (capital in the words of the reviewer).

At ORNL it is fairly clear that eCO₂ exacerbated resource limitation rather than relieving it. At Duke, LAI increased early in the experiment, and fine-root mass continued to increase through most of the experiment. However, the potential gains in resource acquisition from these biomass trends were not evident as trends in NPP. What’s interesting at Duke is that LAI appears to have somewhat stabilized (little trend but substantial variability) in both treatments and that value around which they have stabilized is higher under eCO₂. This higher value suggests that eCO₂ is not simply accelerating development but is facilitating consistently higher values of LAI. Also, LAI was not

solely measured using optical techniques, the primary measurement was from litter-traps, see McCarthy et al., (2010) and Table 4 in our manuscript.

At Rhinelander LAI does not stabilize over the course of the experiment, and LAI is on a higher linear trend (higher intercept, similar slope) under eCO₂ compared against ambient. So it does appear that eCO₂ has advanced the developmental stage of the stand. Though there is no evidence for exponential growth. We characterize Rhinelander as in the reorganising stage, and in a state of expanding resource acquisition.

193 The net biomass increment...

It is the biomass increment over the time period. We're not sure what would be net and what would be gross biomass increment. Gross would be the cumulative NPP would it not? We have made no changes.

193-194 That sentence simply says that all 4 stands grew.... Who would have expected otherwise?

Yes, that was somewhat stating the obvious! We have deleted the sentence.

195 I do understand why you coined Cinc-veg. Yet, the difference to the popular NEP is not made obvious to readers. The main reason is that you exclude soil C. All your calculations are built upon growth and biometry data.... You could employ NEP with a caveat in methods... Would make reading so much easier... Maybe NEP_v could do? Just an idea...

We introduced Cinc_{veg} as C_{veg} and C_{soil} are relatively commonly used and we wanted to emphasise that this variable is a change in the C stock over a multi-annual time-period. Furthermore, NEP refers to ecosystem-scale measurements, which we are not discussing. Considering the reviewer's comment we have revised Cinc_{veg} to ΔC_{veg} which is a more appropriate term.

204 I like that clear statement

Thanks, we have tried to be as clear elsewhere in the revised manuscript.

217 time? ...perhaps duration, time laps, period?

We have replaced with "**period over which**"

218 On longer time scales, vegetation turnover is determined....

We have rephrased the sentence (lns 228-230):

Vegetation turnover is determined by allocation of NPP among tissues with differing turnover rates (i.e. leaves and fine-roots vs wood), and over longer timescales, tree mortality.

232 ...a biomass response to CO2 enrichment the neg...

Done.

238-239 Couldn't this be species specific? Some taxa are more 'woody' others more 'weedy'. Also wood density and thus, tissue duration differs (think of scrub oak and aspen)

Apologies but it is not totally clear how the reviewer's suggestion relates specifically to the text on lines 238/39. We agree that there are more woody and more weedy species and that the relationship of wood allocation to cNPP could vary for different species. That's why we investigated the relationship.

258 ...a biomass response to CO2, Rhine... I think the FW story adds a good deal of basic science to the analysis that goes beyond the CO2 fertilization issue.

Done. Thanks, our goal is to investigate which processes are key in the observed response rather than just saying that there was a response.

320 please replace 'forest'. KSC is scrub as the name says. This scrub system simply does not deserve the word forest, nor do the young stages of the aspen experiment.

Done.

328 ...evidence that these early...

See below.

329 ...increment were carbon....

Done.

I suggest to be careful with the generalization given the very special settings of these different experiments.

The mixed model analysis actually gives a population-level inference rather than just the sites we are investigating. We understand that a sample across four sites is small, but it is the only sample we have. We have deleted the word "strong" given the small sample size.

339 my comment to line 181 applies here. Had these 29% already been established very early in the experiment and just be retained? Can you bring a bit light into that 'compound interest' story?

The reviewer seems to be talking about two separate things – one a story of compound interest (described and responded to above), and the other a story of initial gains early in the experiment being preserved and making up the majority of the gains seen at the end of the experiment.

From Figure 1 a-d it is clear that for NPP at Rhinelander, Duke, and KSC (all with a biomass response over the full duration of the experiments), the gains are not concentrated in the early few years of the experiment. Duke and Rhinelander have consistently higher NPP under eCO₂ during the whole experiment and the trend analysis shows no interaction of CO₂ with the trend. At KSC CO₂ did not affect the trend and NPP gains under eCO₂ followed the disturbance (Hungate, et al. 2016). At ORNL we agree with the reviewer that gains in NPP decreased towards the end of the experiment, but the gains were across the first 7-8 years rather than the initial few years.

The Figure below shows total biomass over the course of the experiments. The Figure demonstrates a steady increase in the biomass increment over the course of the Rhinelander and Duke experiments and increases that follow disturbance events (beginning of experiment and year 108 post-establishment) at KSC, following NPP as described above.

347 the central role of the nutrient cycle for the C cycle was recently shown for grassland by Reich et al in their April Science article.

We appreciate the reference (which was published ten days after our previous submission) but there are plenty of references for forests, and for the four specific studies analysed. The point made in the text is specific to these four studies and so we stick with our original references.

357-360 Isn't that a repetition?

Agreed, thanks. We have removed this particular sentence. Moreover, we have streamlined the entire discussion to flow more logically, to avoid repeating presentation of results and to focus on interpretation, and to avoid the potential for repetition. Please see revised discussion.

363 NPP is expressed per unit land area. Hence regrowth or seedling/cutting growth from near to nothing such as at KSC and RHINE would yield very little NPP at very high individual growth rates till canopy closure.

This depends on how tightly spaced the individuals were at planting and on the capability of individuals to resprout. As shown definitively in Figure 1, NPP (expressed per unit land area) is highest at KSC in the first year.

I rather think this is related to stand age. Once leaves and fine roots 'filled' the space, there is only wood increment left (both below and above ground) to incorporate more C in biomass per unit land area. It needs no 'strategy' in the sense of a 'priority' to explain that. This is the very nature of steady state (since coined by Odum).

We see the reviewer's point. We have rephrased the statement to read (lns 362-365):

A simple hypothesis for the linear relationship between fW and cNPP (Table 2) is that allocation to resource acquisition organs (i.e. leaves and fine roots) at the decadal scale is fairly well conserved in these even aged stands and that variability in NPP is primarily driven by variability in wood production.

392-407 Since the nutrient availability most likely controlled C incorporation at all 4 sites, it is unfortunate that such nutrient data are not provided. Whatever modelers assumed in that respect will have driven model output.

The models that explicitly represent N cycling do so in many different ways. The only external inputs to the models are from N deposition fields that we took from (Dentener et al., 2006) and measured data at the site. Each model then calculates soil N stocks and mineralisation rates based on their own conceptualizations of all the relevant processes and a 'spin-up' to steady-state prior to the industrial revolution. For a detailed expose of how these processes are represented by models and their consequences for simulating CO₂ responses see the references cited in the manuscript (De Kauwe et al., 2014; Walker et al., 2014; Zaehle et al., 2014).

402 ...models with an N cycle.... based on which input data? So, some of the models employed had no nutrient cycle? How can growth be modeled without nutrients?

These are well established ecosystem models, some of which do not consider explicit N or other nutrient cycles. Some components of nutrient limitation may be simply or empirically represented or parameterised. In any case, these light, C, water, temperature, and VPD only models are being used and have been to predict CO₂ responses in many studies. Only three models from the ensemble do not explicitly represent nutrient cycling and we think it is an interesting point that these do not obviously predict a different CO₂ response to the models that do represent N cycling.

406 refer to Reich et al Science 2018, they showed exactly this.

As above, we prefer to cite literature that is focused on woody ecosystems, rather than a single herbaceous system as studied by Reich et al. We are happy with the six citations we already use to support this statement.

426 However.... suggest to remove that sentence. The experiments (and models) presented here have the same limitation. These central European forests are likely to operate at N saturation given the enormous N-deposition rates as was suggested by Schleppi et al. (2012) in Global Change Biology.

We have revised this paragraph. The line referred to by the reviewer has now been replaced and the following sentence on identification of limiting resources has been removed, (lns 442-444):

These results suggest that in these later successional forests, perhaps in the transition phase, aboveground tree biomass increment was not carbon limited.

Why not refer to Sune Linders OTC experiment with tall spruce? They showed that the CO₂ response could be switched on or off by nutrient addition.

Happy to. The additional text reads (lns 439-441):

At Flakaliden, Sweden, biomass increment in individual trees of Norway Spruce was increased by CO₂ enrichment only when nutrients were also added (Sigurdsson, et al., 2013).

Sigurdsson BD, Medhurst JL, Wallin G, Eggertsson O, Linder S (2013) Growth of mature boreal Norway spruce was not affected by elevated CO₂ and/or air temperature unless nutrient availability was improved. Tree Physiology 33:1192-1205

438 ...NPP with N at our test sites.

Done.

452-454 I find the handling of this matter imbalanced. Ref 54 and 55 provide info on undisturbed systems, 57 is a model with an emphasis on simply showing the importance of mortality and NOT a vigor-lifespan relationship. 58 and 59 talk about fire driven systems, where fast early growth creates a protective bark that better resists fire. These papers do not refer to biological life expectancy. Sure, all this matters, but this § reflects bias ('a broader body'). I do not see that 'broader body' for the current question.

We disagree. 57 (Bigler and Bugmann, 2003) is not just “a model,” the study presents a statistical model of mortality fitted to tree-ring data collected at three sites in Switzerland that were: “mature near-natural or primeval forests with *P. abies* as the dominant tree species” (Bigler and Bugmann, 2003). Their study was intended to “provide insight into growth–mortality relationships and the effect of competition on current growth.” A key conclusion from the study is that inclusion of growth rates in the statistical model improves model classification for dead vs alive trees, but the

association is between low growth rates and increased mortality. We concede that they do not formally analyse the influence of early growth rates, but a comparison of Figures 3 and 4 in that paper show no clear higher early growth in trees that were dead. Similar methods and conclusions can be drawn from 56 and (Bigler and Bugmann, 2004).

With regards to 58 and 59, is fire not an agent of tree mortality?

The most extensive study we could find that addresses mortality and the relationship of (early) growth and mortality that we were able to find in the literature is 61 (>7000 individual trees across 140 sites). 61 found no negative relationship between early growth rates and mortality. Though they do suggest that further work is needed.

We have edited the relevant text to read (ln 459): **there is also a substantial body of evidence.**

455 mortality to increase soil C inputs? Not necessarily. The effect could be neutral or even feeding back negatively on growth because of nutrient trapping in debris.

The full quote from our original manuscript was:

If higher growth rates were to increase mortality rates, soil C inputs would be increased. How soil C responds to CO₂ enrichment is an active area of research that must also be considered in analyses of feedbacks between the atmosphere and terrestrial ecosystems⁵.

We feel this is a reasonable statement. Increased vegetation mortality initially must increase vegetation soil C inputs. To trap additional nutrient as suggested by the reviewer then soil C would likely have to increase. Over the long term “nutrient trapping” could reduce nutrient losses from the system and increase the nutrient capital of the system. If this nutrient capital is not as mineralizable, reducing mineralisation rates, and slowing plant growth then according to the reviewers’ vigor-mortality hypothesis then mortality would decrease. We prefer to avoid these stories of potential feedbacks and stick with our original statement of immediate effects of increased mortality. The revised statement reads (lns 464-467):

If increased growth rates do lead to increased mortality, the immediate consequence will be increased inputs of C to the soil. How soil C responds to CO₂ enrichment is an active area of research that must also be considered in analyses of feedbacks between the atmosphere and terrestrial ecosystems^{5,37}.

All these are minor issues of wording and literature treatment. My comment to line 181 and 339 may deserve a few remarks/numbers, given the systems differed strongly in that respect. The response during this initial stage may, in fact, hold much of the answer to the final outcome.

Please see all of our responses described above.

Reviewer #2 (Remarks to the Author):

The authors have worked diligently to address the reviewers' comments. Thank you! A few questions remain (or came up with the reading of the revised manuscript).

Thanks, we appreciate the recognition of the work we put into the revisions.

Interest to others in community and wider field: The findings are definitely in the interest of the community and in the general field of climate change and global biogeochemistry. It summarizes nicely and in a fairly comprehensive way the main results of FACE experiments after a decade or so of experimentation. I also think the way the used statistical model is state of the art and the inclusion of models tasks to quantify those feedbacks for future climate feedback enhances the discussion. While I am definitely interested to see this work published, I cannot make up my mind whether this is sufficiently novel for a much broader community of natural sciences.

We appreciate the reviewers interest in our work. We feel our work as described by the reviewer is well within the "Aims and Scope" of Nature Communications (taken from the Nature Communications website, <https://www.nature.com/ncomms/about>) and we hope the editor agrees.

Main result reporting: I am wondering whether the percentage NPP response is actual the right metrics to report in the abstract and as a main conclusion. The work suggests that the baseline NPP is shifting, while the response to higher CO₂ (in absolute terms) is much more indicative. Table 2 suggest that across all sites the NPP response is 0.164 ± 0.031 (kg m⁻²). If then expressed as a percentage, it is then divided by the random effect of each site, diluting the precision.

Not exactly. The mixed-effects model provides a population level estimate of the fixed effect, so the percentage reported in the abstract is an estimate of the population response. Yes, the uncertainty is larger than in previous studies, because of the variability across sites as pointed out by the reviewer. In using the site variation to estimate the variance of the population, the mixed-models allows a broader inference space indicated by the larger uncertainty. That said, we think that the absolute response is more relevant to this study than percentage responses and we have reorganised the presentation of results in the abstract and other places to always lead with the absolute response.

New discussion on forest development: I appreciate the inclusion and the discussion of the stages of regrowth in these sites. I am wondering, though, whether they could be even more integrated, by for example discussing the allocation to wood across the different sites. Is there an expectation that a shift from one phase to the next also leads to a shift in wood allocation? Or alternatively, is there a potential change in retention rate along the successional development stages?

Yes, this is the implication at ORNL, i.e. that during the shift to the transition stage of succession there is a shift in wood allocation at the stand scale. We recognise that we could have done a better job of integrating the successional stage in the discussion. We have now revised and added to the discussion of experimental results (lns 367-398):

Random effects in the best statistical model of ΔC_{veg} suggest site-level differences in the biomass retention rate. Rhinelander, Duke, and KSC showed a positive biomass retention rate and thus a decadal biomass response. These three sites were in the reorganising and aggrading phases of secondary succession, with various degrees of expanding above-and-below-ground resource acquisition volumes and thus coupling to resource availability. At these sites, the biomass retention rate was tied via Eq 1 to cNPP and wood allocation. Both the baseline wood allocation fraction (fW_a) and the unit change in fW for a unit change in cNPP (fW_b ; which was conserved across sites) were important (Table 3). Rhinelander and Duke shared a similar fW_a and the difference in cNPP at the sites determined the different biomass retention rates via the second term in Eq 1 (Table 3). fW_a was lower at KSC, leading to a lower biomass retention rate.

At ORNL, the biomass retention rate of 0.144 (-0.093–0.678, 95 % CI) was not statistically different from zero and therefore there was no relationship between ΔC_{veg} and cNPP. The absence of a relationship was not a response to CO₂ enrichment. ΔC_{veg} at ORNL was simply more variable within CO₂ treatments than across treatments, related to high within-treatment variability in fW (Figure 3) that resulted in a highly uncertain biomass retention rate. In several years of the ORNL experiment, annual root production was stimulated by CO₂ enrichment³⁶. However, at the decadal scale, a treatment effect on allocation was not detected (Table S5 & S6). At ORNL, the peak and later decline in NPP in both treatments (Figure 1) was attributed to progressive nitrogen limitation, which was intensified by CO₂ enrichment^{13,24}. In this state of tightening resource availability ΔC_{veg} was controlled by within-treatment variability in fW that was unrelated to variability in cNPP. We propose the hypothesis that cross-plot variability in the timing and intensity of competition as the plots at ORNL shifted from the aggrading phase into the transition phase of secondary succession was the cause of the within-treatment variability in fW. As with the sites in the earlier stages of succession, understanding the ambient condition and successional status is important for interpreting the (lack of) ΔC_{veg} response to CO₂ at ORNL.

Cinc/NPP relationship: I understand that the fraction allocated to wood is a modeled variable as it relates Cinc with NPP. However, I think that in each of the sites, wood increment has been measured in order to obtain NPP. Can the estimated allocation to wood via Cinc-NPP relationship be verified by actually calculating the 'real' (i.e. estimated from wood increment) data? There may be a need to make certain assumption (no mortality, for example), but I don't think these assumptions would be more tenuous than the empirical fW (Cinc/NPP) function in the existing manuscript?

In the analysis, fW is calculated from measurements of wood increment divided by NPP (which is calculated as the sum of measured growth increments off all tissue types; see Table 4), as suggested by the reviewer. We have verified the Cinc-NPP relationship with the fW analysis. Apologies if we are missing something. We have added some text to the methods to be clear that allocation fractions were calculated from measurements (lns 524-528):

All components of NPP and biomass were measured, though somewhat different methods were used at each site. For our analyses, allocation fractions were calculated as the organ production divided by total NPP (which was calculated as the sum of production of all organs: leaves, wood, coarse-roots, fine-roots). Details of how the measurements were made can be found in Table 4 and the references cited therein.

Model data discrepancy and its discussion: It becomes clear that one of the failures of models to capture the CO₂ effect is their failure to predict wood allocation at these sites. Many models currently have very simple rules for wood allocation (fixed, pipe model, and others). However, it is less clear how models can improve. As I understand fWa has site specific random effects. Yet models need to be able to predict these in order to capture the CO₂ effect. I would appreciate potential discussion points (does the literature suggest some relationship? - Is this something that needs further evaluation and what kind of data could help with that?).

Thanks, this is a great question. We have now added the following text to the discussion (416-421):

Overall, our results suggest that allocation rules were more constrained across sites than across models, though successional stage and disturbance regime did drive cross-site differences in allocation. Models with allometric constraints, such as the pipe model ³⁸, tended to perform better in a previous analysis ³² and a representation of tree size and potentially forest structure through succession may help models to better implement the more conservative allometric constraints implied by the observations and previous analyses ^{26,39}.

Minor and editorial stuff:

L147 I believe it should be N^{15} instead of N_{15}

Thanks, done.

L157 What does "this value" refer to? Resource acquisition phase, saturation?
Please clarify

Modified to: **this saturation value**

L158 "However, this evidence for expanding resource acquisition volumes was not apparent in the NPP data". How would expanding resource acquisition present itself in the NPP data?

Yes, this was not clear. We would expect increasing trends in resource acquisition volumes to result in increases in resource acquisition, which should result in increases in NPP. However there is no trend in NPP data at Duke. We modify the sentence as follows (ln180-181):

However, the lack of a trend in NPP data suggests that the expanding resource acquisition volumes over time were not yielding increased resource acquisition (Figure 1b).

L163: "the trend was highly variable" - meaning no trend?

Thanks, we have modified the sentence as follows: **no clear trend was apparent**

L236: Isn't 3) a logical consequence of 1) and 2)? That is the third prediction is not independent? A simple therefore before "3)" might just be enough to clarify.

Yes, exactly, thanks. We have modified the sentence as suggested.

Reviewer #3 (Remarks to the Author):

The authors' responses to previous comments are generally on target, and appreciated. With the improvements, it is now easier to see the issues that need attention.

The importance of the comparability of the data cannot be emphasized enough when only four sites are available. It is not necessary to include in the main text the entire Table 4 from the supplementary information, but it is important to state in a paragraph or so whether the way these data were collected/measured may influence the results. Table 4 is somewhat difficult to read, but it looks like coarse roots are generally calculated at all sites as a function of diameter-based allometric relationships. KSC looks to have had more attention to the root-related estimates, with both soil cores and mini-rhizotrons, and an isotopic tracer approach was used for a turnover rate. How do authors know that differences between the different pools on these sites, especially roots, is not largely related to the methods for producing the data/estimates? A summary paragraph or two in the text to convince the reader the data from these sites is comparable in spite of different collection techniques and in spite of perhaps using more localized biomass equations would be useful.

We agree that the variability in measurements across the different sites are likely to influence the results to some degree. The use of mixed-models allows for site-level differences in measurements technique to be accounted for. Thus the variability caused by method differences across sites, as well as other artifactual and real variability across sites, is reflected in the uncertainty of the population-level parameters. It is analogous to a block design where we know there is variability among blocks, but are unable to fully describe that variability in the fixed effects of our statistical model. These measurement differences are not accounted for in the mixed-effect model when sites are compared with one another; however, we believe the influence of these measurements differences to be small. This we describe in the methods text (lns 544-573):

An issue in all meta-analysis type studies is that variability in the way that measurements have been made may result in artifacts that obscure true differences across experiments. Random effects in mixed models account for unidentified variability across samples within a group when calculating population level fixed-effects, but cannot account for unidentified variability when specific sample-to-sample comparisons are made. The decadal biomass increment is mostly a result of wood increment, which is robust across sites as described above. Thus the primary result of a $1.05 \pm 0.26 \text{ kg C m}^{-2}$ stimulation of biomass increment by CO_2 enrichment is robust. Measurements within experiment sites were consistent, which means that our second major result that the biomass retention rate is independent of CO_2 is robust. Measurements of fine-root production are likely to vary the most among sites, though each method was state-of-the-art, and are thus most likely to influence cross-site comparisons. For example, at KSC fine-root biomass was quantified with a combination of cores and mini-rhizotrons, and coarse roots with a combination of allometry and ground-penetrating radar. Although the same suite of methods was not used at other sites, we have no reason to believe such differences explain 5-10 times higher root biomass at the KSC site. Furthermore, at two sites (ORNL and KSC) with different methods of measuring root production, and largely different values of fine-root production, the measurements of production both under ambient CO_2 and in response to CO_2 enrichment were supported by measurements of fine-root biomass during final harvests^{67,68}. Thus while the cross-site comparisons of the exact value of

the biomass retention rate may be coloured by methodological differences, we expect these biases to be small.

KSC used biomass increment and litterfall to calculate NPP⁶⁸. In a few years this calculation resulted in negative NPP in some plots because biomass (in particular, fine-root biomass; Figure 11) fluctuated strongly year-to-year but calculations of litterfall did not fluctuate as strongly. Based on the conceptualisation of NPP used at the other three sites (i.e. the sum of gross production), negative values of NPP at KSC mean that components of either litterfall or biomass were not measured. Negative NPP and not knowing whether the unmeasured litterfall or biomass component of NPP varies annually or not makes annual time-scale comparisons with the other three sites difficult. At multi-annual timescales the large fluctuations in biomass that cannot be accounted for with litterfall measurements are averaged leaving a single directional bias, a bias that is likely present and similar in both cumulative NPP and biomass.

We are not 100 % sure what the reviewer means by “in spite of using more localized biomass equations.” Localised allometric equations were derived to calculate wood biomass for the species’ present at each site. I think that we agree with the reviewer that this makes the data more accurate, and thus more comparable.

Part of the reason we only have four sites to analyse is that we had strict selection criteria that were designed specifically so that the sites in the analysis were comparable. We have already excluded many studies that measured only above-ground elements of production and biomass, or were measured differently, or at different scales. Our selection criteria were:

- * Lasting a decade or longer
- * Ecosystem scale measurements
- * Measurements that quantified all elements (leaf, stem, coarse-root, and fine-root) of production and biomass

These are the best only data that we have given our our strict selection criteria that was intended to minimise methodological differences (e.g. we do not include pot studies nor studies that only measure above-ground production biomass).

We prefer to keep Table 4 in the main text.

And move Table 4 back to supplemental, and reference that it was there.

We prefer to keep Table 4 along with the text that describes it in the methods.

Estimating NPP from above-ground inventory data is an accepted technique (see Jenkins et al 2001 for example). A study like that one makes clear how much NPP is related to biomass increment by definition.

We are investigating total biomass, not just above-ground. From our interpretation of Jenkins, they show that above-ground production and biomass are related, but not by definition. We do not agree with the premise that NPP is related to biomass increment by definition, especially when total (above and below ground) NPP and biomass is considered at the decadal scale. This is clear in our data that show no relationship between biomass increment and NPP at ORNL.

If the biomass increment to NPP lay on the 1:1 line in Figure 2 then we would agree with the premise that biomass increment and NPP are related by definition, but they do not fall on the 1:1.

We now reference the still relevant statement from Strain and Bazzaz 1983 (lns 87-91):

“ ... the initial effect of elevated CO₂ will be to increase NPP in most plant communities. This increase in NPP could be limited or reversed by nutrient availability, herbivory, or successional dynamics [...]. However, even without such effects, a critical question is the extent to which the increase in NPP will lead to a substantial increase in plant biomass. Alternatively, increased NPP could simply increase the rate of turnover of leaves or roots without changing plant biomass.”

Here, the wording of the research questions need to be reconsidered, especially the question “is there a relationship between biomass increment and cumulative NPP”? This an obvious answer: yes, by definition.

See our statement above. At the decadal scale especially, NPP and biomass increment are not related by definition. The lack of a relationship at ORNL proves that NPP and biomass are not related by definition.

However, we agree with the reviewer that our research questions could be stated more clearly. See our response below.

Mortality will affect standing live biomass increment. If one is examining a graph of only biomass increment and only cumulative NPP plotted against each other those variables may seem unrelated, but it is just that mortality is being ignored. Consider modifying the questions for clarification. The abstract indicates one question is: is the relationship of wood allocation to NPP affected by atmospheric CO₂ level? Is this one of the questions? It seems as if I am still reading an early version of a manuscript, and that the ideas in the text need to be made more clear and crisp.

We agree that our objectives as stated were somewhat confusing. We have now stated more clearly the three primary objectives of the study in the introduction (lns 115-119):

Our objectives are three-fold: 1) Determine whether a decade of CO₂ enrichment in woody ecosystems leads to an increase in the vegetation biomass increment (ΔC_{veg}), 2) Interpret any observed biomass response through the effects of CO₂-enrichment on NPP and carbon allocation, and 3) Evaluate the ability of an ensemble of terrestrial ecosystem models, commonly used to predict vegetation responses to CO₂, to reproduce the observed responses.

Also we have substantially rearranged the abstract, and initial paragraphs results and discussion to put biomass increment results up front. Followed by NPP and allocation results to interpret the biomass increment results.

Another possible issue and why a focus on data is important:if CO₂ affects tree growth such that the taper of the tree bole is affected, such as, trees have a larger diameter but do not proportionately increase in height or proportionately do not increase in diameter at different heights up the tree, then using the same biomass models for different time periods would ignore that effect. The

independent relationship of wood allocation to NPP may be masked without having more precise estimates of wood production. (an example related to crown dynamics is Valentine et al. (2013)).

FACE researchers were very aware of the possibility of size and eCO₂ altering allometry such that biomass estimates based solely on diameter might not be appropriate. Hence, at ORNL, the initial allometric equation included a taper function and wood density (Norby et al., 2001), both of which were monitored throughout the experiment. At the end of the experiment, this initial allometry was found to remain valid based on whole tree/root harvesting from each plot. At Duke FACE, there was a similar recognition of the possibility of the height-diameter relationship changing, leading to a reanalysis of biomass estimates (McCarthy et al., 2010).

Turning to the ecosystem models part of the study: It is not clear why it is included here. Lines 278-279 indicate that they were used "to simulate these four CO₂ enrichment experiments and identify any common areas where models might be improved." There is no listing of common areas where models might be improved.

A listing of common areas was included in the discussion, along with a more general paragraph on the model failures in N cycle representation. The slightly edited text now reads (previous draft lns 392-406, current submitted draft lns 403-424):

We highlight four findings related to C allocation that will help models to improve simulated ΔC_{veg} responses to CO₂ enrichment: 1) across CO₂ treatments, fW was a linear function of cNPP; 2) large variability in the predicted intercept of the relationship (fW_a) led to large variability in the predicted biomass retention rate; 3) model predictions of the wood allocation response to cNPP (fW_b) were low biased; and 4) models did not capture the low fW_a at KSC that is assumed an adaptation to frequent disturbance. Models vary substantially in how C allocation is implemented resulting in substantial model C sink variability^{32,37}. Overall, our results suggest that allocation rules were more constrained across sites than across models, though successional stage and disturbance regime did drive cross-site differences in allocation. Models with allometric constraints, such as the pipe model³⁸, tended to perform better in a previous analysis³² and a representation of tree size and potentially forest structure through succession may help models to better implement the more conservative allometric constraints implied by the observations and previous analyses^{26,39}.

In these four ecosystems, the N constraints on NPP responses to elevated CO₂ were met by increased N uptake, rather than an increase in N use efficiency^{31,40}. In the models with an N cycle, under-prediction of the cNPP response at Rhinelander and Duke was likely a result of overly strong N constraints that did not allow flexibility in the coupling of the C and N cycles^{31,40,41}. Understanding the coupling of the C and N cycles through plant-microbe C and N dynamics and the C cost associated with N uptake will help to improve model simulations and is an ongoing area of research^{42,43,31,44-46}. Furthermore, the strong nutrient constraint at ORNL, imposed by stand development, and within-treatment variability in allocation patterns makes a case for representing succession tied to dynamics of resource limitation in models.

Examining ensemble results usually is not that helpful when considering model improvement because the results are all muddled.

We have performed a number of studies in the past that look in detail at individual model results and how they compare to data (e.g. De Kauwe et al., 2014, 2013; Walker et al., 2014; Zaehle et al.,

2014). These studies took an extremely detailed look at various model assumptions, explaining how particular assumptions and processes lead to divergent results. These papers were well received yet are most often cited for results that pertained to the ensemble as a whole. For example, almost all models predicted and increase in NUE rather than N uptake at Duke and ORNL and that is why they were unable to sustain the observed CO₂ response (Zaehle et al., 2014). When analysing our model results, we found that the results from individual models were various (muddled as pointed out by the reviewer) which were difficult to synthesise into a clear story and biased the paper towards explaining individual models. We wanted to have a balance among our three main objectives and found that analysing the properties of the model ensemble as a whole was fruitful, resulting in some clear areas where all models were failing.

Instead (back to lines 320-321) the modeling seems to be included to investigate ecosystem scale effects. For example, as presented in lines 288-290 ("Partitioning the modelled Cincveg response to CO₂ enrichment into the cNPP response to CO₂ and the CO₂-independent biomass retention rate, as described for the observations, allows us to identify which of these processes was responsible for the model bias."), it looks like the models are being examined to test how they assume partitioning occurs, and comparing results to these data, which includes some modeled pools or pools estimated using different methods so back to the importance of the assumption of data comparability. Perhaps the main confusion is the text (lines 278-279) that indicates "any common areas" for improvement are doing to be identified. It sounds as if the reader should expect a list. But the models appear to be introduced to investigate a specific phenomenon.

With regards to a list, please see our statement above. The models are used to evaluate whether they can reproduce the observations which is important because many of these models are used in ensembles run by the Global Carbon Project and Climate Model Intercomparison Project. When analysing the observations we decomposed the empirical relationships into a conceptual framework with components that represent different processes. This allowed some insight into the different processes leading to the observed CO₂ response of biomass increment. We then used the various elements of the decomposition and evaluated the model against them. This decomposition method was applied successfully in a number of our previous studies (De Kauwe et al., 2013; Walker et al., 2015; Zaehle et al., 2014) and for an overall summary description of the method and findings see Medlyn et al., (2015). The decomposition we employed in the current study is for the same purpose, though evaluating the models in a way that relates to the various processes that can contribute to a CO₂ response of biomass increment.

Over the years, the FACE program studies have focused on crucial questions, and as a result are of interest and importance in the scientific literature. Strictly from a research and a statistical point of view, in general, making inferences based on four ecosystem sites to address questions on effects of CO₂ to apply to all temperate woody ecosystems seems like a stretch. Lines 320-321 says "The scale of these FACE and OTC studies (ecosystem and decadal scales) allows analysis of forest responses to CO₂ enrichment at scales at which the carbon cycle becomes relevant to climate change." It is a stretch to have one data point (meaning the four sites each cover one decade) and say the results apply to decadal scales. Nonetheless, this publication would be typical for FACE studies given there are no alternative sources of data, and the results may be of interest and be very meaningful to that scientific community.

Agreed, but these are the data we have. In an ideal world, we would have FACE and OTC experiment replicated in all of the world's major biomes and climatic regions, and at various stages of succession. That would require a multi-billion-dollar research program in a global research climate that is not very favourable to FACE experiments.

By dictionary definition, 'decadal' refers to both a single decade and multiple decades. We now make it clear in the text that we mean a single decade when using the term 'decadal' (lns 472-475):

However, four sites is a very small sample size of the temperate woody ecosystem population and a single decade is at the lowest end of the decadal scale. The successional context, and the dynamics of limiting resources through successional stages, gives us a framework to scale CO₂ responses to greater spatial and temporal scales.

The text needs tightening up though because there appear to be inconsistencies throughout as to what the questions are being investigated (see abstract—are all the questions answered directly there?), and why certain methods are used. The manuscript needs more thoughtful attention to be understood by the broader scientific community.

We have edited the final two (now three) paragraphs of the introduction to separate the three main objectives from the tasks and analyses employed to achieve the objectives (lns 115-139):

Our objectives are three-fold: 1) Determine whether a decade of CO₂ enrichment in woody ecosystems leads to an increase in the vegetation biomass increment (ΔC_{veg}), 2) Interpret any observed biomass response through the effects of CO₂-enrichment on NPP and carbon allocation, and 3) Evaluate the ability of an ensemble of terrestrial ecosystem models, commonly used to predict vegetation responses to CO₂, to reproduce the observed responses.

The shifts in resource availability associated with secondary succession¹⁷⁻¹⁹ are likely key factors influencing the CO₂ response¹⁸⁻²¹. Therefore, to assess whether successional stage can help to interpret CO₂ responses, we begin by inferring the successional stage of each site from trends in NPP, leaf area, and fine-root biomass. To analyse responses to elevated CO₂ across all four sites in a unified statistical framework, we use linear mixed-effects models treating site as a random effect. Site as a random effect treats each experiment as a sample drawn from a population (in the statistical sense), allowing the estimation of a population-level fixed effect and its associated uncertainty. Thus the mixed-model analysis in this study allows an estimate of the general effect of CO₂ in the population of early secondary-successional, temperate woody ecosystems and the random effects estimate inter-site differences. We use Akaike Information Criterion corrected for finite sample size (AICc) to select the best, most parsimonious, statistical model from a set of candidate models.

A set of equations are derived to decompose the biomass increment response to CO₂ and to interpret the empirical parameters of the linear mixed-effects models in the context of carbon allocation (see Methods for more details). We evaluate the ability of a model ensemble to predict decadal biomass responses to CO₂ enrichment and identify areas general areas of model failure where the whole ensemble fails to reproduce observations. We evaluate the models against the biomass response and the decomposition of the biomass response. Finally, we provide recommendations for future research to help understand and predict vegetation biomass responses to CO₂ enrichment.

Also we have edited the abstract, results, and discussion to be clearer in our approach and be more consistent in the order in which results are presented and discussed. We feel this has greatly improved the logical progression of the manuscript and the ease of interpreting the study. The changes are too substantial to quote completely here, please see the tracked changes version of the revised manuscript.

Jenkins, Birdsey, et al. 2001. Biomass and NPP Estimation for the Mid-Atlantic Region (USA) Using Plot-Level Forest Inventory Data. *Ecological Applications*, 11(4), pp. 1174-1193

Valentine, HT. 2013. Crown-rise and crown-length dynamics: application to loblolly pine. *Forestry*, 86, (3), 371-375

References

- Bigler, C., Bugmann, H., 2004. Predicting the time of tree death using dendrochronological data. *Ecological Applications* 14, 902–914. <https://doi.org/10.1890/03-5011>
- Bigler, C., Bugmann, H., 2003. Growth-dependent tree mortality models based on tree rings. *Can. J. For. Res.* 33, 210–221. <https://doi.org/10.1139/x02-180>
- De Kauwe, M.G., Medlyn, B.E., Zaehle, S., Walker, A.P., Dietze, M.C., Hickler, T., Jain, A.K., Luo, Y., Parton, W.J., Prentice, I.C., Smith, B., Thornton, P.E., Wang, S., Wang, Y.-P., Wårlind, D., Weng, E., Crous, K.Y., Ellsworth, D.S., Hanson, P.J., Seok Kim, H.-, Warren, J.M., Oren, R., Norby, R.J., 2013. Forest water use and water use efficiency at elevated CO₂: a model-data intercomparison at two contrasting temperate forest FACE sites. *Global Change Biology* 19, 1759–1779. <https://doi.org/10.1111/gcb.12164>
- De Kauwe, M.G., Medlyn, B.E., Zaehle, S., Walker, A.P., Dietze, M.C., Wang, Y.-P., Luo, Y., Jain, A.K., El-Masri, B., Hickler, T., Wårlind, D., Weng, E., Parton, W.J., Thornton, P.E., Wang, S., Prentice, I.C., Asao, S., Smith, B., McCarthy, H.R., Iversen, C.M., Hanson, P.J., Warren, J.M., Oren, R., Norby, R.J., 2014. Where does the carbon go? A model–data intercomparison of vegetation carbon allocation and turnover processes at two temperate forest free-air CO₂ enrichment sites. *New Phytol* 203, 883–899. <https://doi.org/10.1111/nph.12847>
- Dentener, F., Drevet, J., Lamarque, J.F., Bey, I., Eickhout, B., Fiore, A.M., Hauglustaine, D., Horowitz, L.W., Krol, M., Kulshrestha, U.C., Lawrence, M., Galy-Lacaux, C., Rast, S., Shindell, D., Stevenson, D., Van Noije, T., Atherton, C., Bell, N., Bergman, D., Butler, T., Cofala, J., Collins, B., Doherty, R., Ellingsen, K., Galloway, J., Gauss, M., Montanaro, V., Mueller, J.F., Pitari, G., Rodriguez, J., Sanderson, M., Solomon, F., Strahan, S., Schultz, M., Sudo, K., Szopa, S., Wild, O., 2006. Nitrogen and sulfur deposition on regional and global scales: A multimodel evaluation. *Glob. Biogeochem. Cycle* 20. <https://doi.org/10.1029/2005GB002672>
- McCarthy, H.R., Oren, R., Johnsen, K.H., Gallet-Budynek, A., Pritchard, S.G., Cook, C.W., LaDeau, S.L., Jackson, R.B., Finzi, A.C., 2010. Re-assessment of plant carbon dynamics at the Duke free-air CO₂ enrichment site: interactions of atmospheric CO₂ with nitrogen and water availability over stand development. *New Phytologist* 185, 514–528. <https://doi.org/10.1111/j.1469-8137.2009.03078.x>
- Medlyn, B.E., Zaehle, S., De Kauwe, M.G., Walker, A.P., Dietze, M.C., Hanson, P.J., Hickler, T., Jain, A.K., Luo, Y., Parton, W., Prentice, I.C., Thornton, P.E., Wang, S., Wang, Y.-P., Weng, E., Iversen, C.M., McCarthy, H.R., Warren, J.M., Oren, R., Norby, R.J., 2015. Using

- ecosystem experiments to improve vegetation models. *Nature Clim. Change* 5, 528–534. <https://doi.org/10.1038/nclimate2621>
- Norby, R.J., Gunderson, C.A., Wullschleger, S.D., O’Neill, E.G., McCracken, M.K., 1992. Productivity and compensatory responses of yellow-poplar trees in elevated CO₂. *Nature* 357, 322–324. <https://doi.org/10.1038/357322a0>
- Norby, R.J., Todd, D.E., Fults, J., Johnson, D.W., 2001. Allometric determination of tree growth in a CO₂-enriched sweetgum stand. *New Phytologist* 150, 477–487. <https://doi.org/10.1046/j.1469-8137.2001.00099.x>
- Norby, R.J., Wullschleger, S.D., Gunderson, C.A., Johnson, D.W., Ceulemans, R., 1999. Tree responses to rising CO₂ in field experiments: implications for the future forest. *Plant, Cell & Environment* 22, 683–714. <https://doi.org/10.1046/j.1365-3040.1999.00391.x>
- Norby, R.J., Wullschleger, S.D., Gunderson, C.A., Nietch, C.T., 1995. Increased growth efficiency of *Quercus alba* trees in a CO₂-enriched atmosphere. *New Phytologist* 131, 91–97. <https://doi.org/10.1111/j.1469-8137.1995.tb03058.x>
- Walker, A.P., Hanson, P.J., De Kauwe, M.G., Medlyn, B.E., Zaehle, S., Asao, S., Dietze, M., Hickler, T., Huntingford, C., Iversen, C.M., Jain, A., Lomas, M., Luo, Y., Mccarthy, H., Parton, W.J., Prentice, I.C., Thornton, P.E., Wang, S., Wang, Y.-P., Wårlind, D., Weng, E., Warren, J.M., Woodward, F.I., Oren, R., Norby, R.J., 2014. Comprehensive ecosystem model-data synthesis using multiple data sets at two temperate forest free-air CO₂ enrichment experiments: Model performance at ambient CO₂ concentration. *J. Geophys. Res. Biogeosci.* 119, 937–964. <https://doi.org/10.1002/2013JG002553>
- Walker, A.P., Zaehle, S., Medlyn, B.E., De Kauwe, M.G., Asao, S., Hickler, T., Parton, W., Ricciuto, D.M., Wang, Y.-P., Wårlind, D., Norby, R.J., 2015. Predicting long-term carbon sequestration in response to CO₂ enrichment: How and why do current ecosystem models differ? *Global Biogeochem. Cycles* 2014GB004995. <https://doi.org/10.1002/2014GB004995>
- Zaehle, S., Medlyn, B.E., De Kauwe, M.G., Walker, A.P., Dietze, M.C., Hickler, T., Luo, Y., Wang, Y.-P., El-Masri, B., Thornton, P., Jain, A., Wang, S., Wårlind, D., Weng, E., Parton, W., Iversen, C.M., Gallet-Budynek, A., Mccarthy, H., Finzi, A., Hanson, P.J., Prentice, I.C., Oren, R., Norby, R.J., 2014. Evaluation of 11 terrestrial carbon–nitrogen cycle models against observations from two temperate Free-Air CO₂ Enrichment studies. *New Phytol* 202, 803–822. <https://doi.org/10.1111/nph.12697> Enrichment studies. *New Phytol* 202, 803–822. <https://doi.org/10.1111/nph.12697>

REVIEWERS' COMMENTS:

Reviewer #2 (Remarks to the Author):

I have reviewed this work multiple time and it has gotten better each time in terms of being more detailed, working in the special circumstances each of those FACE experiments have been carried out, while maintaining the major messages. The works hows the response to CO₂ being a NPP response to higher CO₂, while wood allocation playing a major role in carbon storage. Also, the drawbacks of current models are discussed to the extent that there remain differences between the sites.

I appreciate the author's hard work on this. I raised earlier doubt about how the work appeals to a general audience, which I still have. But I value very much the research approach and the communications on the findings as someone working in this field. I feel my concerns have been diligently addressed (I also appreciate the responses to my co-reviewers suggestion) and I have no further revision request in this round.